# Statistical distribution of mirror mode-like structures in the magnetosheaths of unmagnetised planets: 2. Venus as observed by the Venus Express spacecraft

Martin Volwerk[1], Cyril Simon Wedlund[1], David Mautner[1], Sebastián Rojas Mata[2], Gabriella Stenberg Wieser[2], Yoshifumi Futaana[2], Christian Mazelle[3], Diana Rojas-Castillo[4], César Bertucci[5], and Magda Delva[1]

[1]Space Research Institute, Austrian Academy of Sciences, Graz, Austria
[2]Swedish Institute of Space Physics, Kiruna, Sweden
[3]Institut de Recherche en Astrophysique et Planétologie (IRAP), Université de Toulouse, CNRS, UPS, CNES, Toulouse, France
[4]Instituto de Geofísica, Universidad Nacional Autónoma de México, Coyoacán, Mexico
[5]Instituto de Astronomía y Física del Espacio, Ciudad Autónoma de Buenos Aires, Argentina

**Correspondence:** Martin Volwerk (martin.volwerk@oeaw.ac.at)

**Abstract.** In this series of papers, we present statistical maps of mirror mode-like (MM) structures in the magnetosheaths of Mars and Venus and calculate the probability of detecting them in spacecraft data. We aim to study and compare them with the same tools and a similar payload at both planets. We consider their dependence on Extreme Ultraviolet (EUV) solar flux levels (high and low).

The detection of these structures is done through magnetic field-only criteria and ambiguous determinations are checked further. In line with many previous studies at Earth, this technique has the advantage of using one instrument (a magnetometer) with good time resolution facilitating comparisons between planetary and cometary environments.

Applied to the magnetometer data of the Venus Express (VEX) spacecraft from May 2006 to November 2014, we detect structures closely resembling MMs lasting in total more than $93,000\,\mathrm{s}$, corresponding to about 0.6% of VEX's total time spent in the Venus's plasma environment. We calculate MM-like occurrences normalised to the spacecraft's residence time during the course of the mission. Detection probabilities are about 10% at most for any given controlling parameter.

In general, MM-like structures appear in two main regions, one behind the shock, the other close to the induced magnetospheric boundary, as expected from theory. For solar maximum, the active region behind the bow shock is further inside the magneosheath, near the solar minimum bow shock location. The ratios of the observations during solar minimum and maximum are slightly dependent on the depth $\Delta B/B$ of the structures, deeper structures are more prevalent at solar maximum. A dependence on solar EUV (F10.7) flux is also present, where at higher F10.7 flux the events occur at higher values than the daily average value of the flux. The main dependence of the MM-like structures is on the condition of the bow shock: for quasi-perpendicular conditions the MM occurrence rate is higher than for quasi-parallel conditions. However, when the shock becomes "too perpendicular" the chance of observing MM-like structures reduces again.

Combining the plasma data from the Ion Mass Analyser with the magnetometer data shows that the instability criterion for MMs is reduced in the two main regions where the structures are measured, whereas it is still enhanced in the region in-between these two regions, implicating that the generation of MMs is transferring energy from the particles to the field. With the addition of the ELectron Spectrometer data, it is possible to show that there is an anti-phase between the magnetic field strength and the density for the MM-like structures.

This study is the second of a series of papers on the magnetosheaths of Mars and Venus.

## 1   Introduction

Mirror modes (MMs) are ubiquitous structures in space plasmas, which consist of trains of magnetic depressions combined with plasma density enhancements in anti-phase. They are stationary in the plasma frame and convect with the plasma flow. Most often, these structures are found in planetary magnetosheaths, behind a quasi-perpendicular bow shock. MMs have been

found at Earth (e.g., Tsurutani et al., 1982; Baumjohann et al., 1999; Lucek et al., 1999a; Soucek et al., 2015), Venus (e.g., Bavassano Cattaneo et al., 1998; Volwerk et al., 2008b, c; Schmid et al., 2014; Volwerk et al., 2016), Mars (e.g., Bertucci et al., 2004; Espley et al., 2004; Simon Wedlund et al., 2022), Jupiter (e.g., Erdös and Balogh, 1996; Joy et al., 2006), Saturn (e.g., Bavassano Cattaneo et al., 1998), and comets (e.g., Mazelle et al., 1991; Glassmeier et al., 1993; Schmid et al., 2014; Volwerk et al., 2014).

### 35   1.1   MM instability and temperature anisotropy

MMs are created by a temperature anisotropy in the plasma, where the perpendicular temperature, $T_\perp$ (with respect to the magnetic field) is higher than the parallel temperature, $T_\parallel$. Hasegawa (1969) showed that for a bi-Maxwellian multi-component plasma the instability criterion is given by:

$$\text{MMI} = 1 + \sum_i \beta_{i\perp} \left( 1 - \frac{T_{i\perp}}{T_{i\parallel}} \right) < 0, \tag{1}$$

where

$$\beta_{i,\perp} = \frac{n_i k_B T_{i\perp}}{B^2/2\mu_0}, \tag{2}$$

is the perpendicular plasma-$\beta$ of species $i$, the ratio of perpendicular (to the magnetic field) plasma pressure and the magnetic pressure. Here $n_i$ is the density of species $i$, $k_B$ is Bolzmann's constant and $\mu_0$ is the permeability of vacuum.

This temperature anisotropy can give rise to two different instabilities: the (Alfvén) ion cyclotron instability for low-$\beta$

plasma, and the mirror mode instability for high-$\beta$ plasma (Gary, 1992; Gary et al., 1993). In this paper we will only consider

the solar wind plasma and, therefore, only $i = p$ (protons). Southwood and Kivelson (1993) rewrote the instability criterion, for protons $p$ only, as $R_{\text{SK}} > 1$, where:

$$R_{\text{SK}} = \frac{T_{p,\perp}/T_{p,\parallel}}{1 + 1/\beta_{p,\perp}} \tag{3}$$

which is often used in papers lately (e.g., Wang et al., 2020), sometimes enhanced through a modified $\beta^*$, which takes into account the ion-Larmor radius effects (Pokhotelov et al., 2004).

The increase in the perpendicular temperature or pressure can be created in various ways, which may be concurrent in the plasma: through pick-up, where the newly created ion starts gyrating around the magnetic field; through perpendicular energization whilst crossing the quasi-perpendicular bow shock; and through slow changes in the magnitude of the magnetic field with conservation of the first adiabatic invariant. The first process will occur mainly in the solar wind interaction with the planetary exosphere (with the exception of Jupiter's magnetosphere and the Galilean moons, where the Jovian corotating magnetic field and magnetospheric plasma is taking the role of the solar wind) in the low-$\beta$ plasma case and generation of ion cyclotron waves will take place (Delva et al., 2008, 2009, 2011, 2015; Schmid et al., 2021). After crossing the quasi-perpendicular bow shock (where the IMF direction is near-perpendicular to the bow shock normal,, with this angle $\theta_{Bn} > 45°$) the anisotropy is increased, as is the plasma-beta and the MM instability will take over. The third process will occur mainly near the magnetic pile-up boundary where the magnetic field gets compressed and slowly increases in strength whilst getting closer to Venus.

The temporal evolution of MMs, while they are convected with the plasma flow, has been discussed by Hasegawa and Tsurutani (2011). It was assumed that there is a Bohm-like diffusion (Bohm et al., 1949) taking place in the MM structures, where the higher frequencies of the structure diffuse faster than the lower ones, and thereby the MMs grow in size. This phenomenon was shown to occur at Venus and at comet 1P/Halley (Schmid et al., 2014).

The temperature anisotropy of the plasma is an important factor in the generation of MMs. Lately, the data of the Ion Mass Analyser (IMA) of the ASPERA-4 instrument (Barabash et al., 2007), as part of the Venus Express mission, have been re-evaluated and reprocessed by Bader et al. (2019), with the special focus of deducing the proton temperatures, $T_\parallel$ and $T_\perp$. This resulted in maps displaying, amongst others, the temperature anisotropy necessary for the MM instability criterion in Eq. (1). It also showed that mainly in the "near-subsolar magnetosheath" there was a large ratio of $T_\perp/T_\parallel \approx 1.56$, whereas in other regions this ratio was $\sim 1$.

Rojas Mata et al. (2022) extended this study and also took into account the possible differences between solar minimum and maximum conditions. They found that $T_\parallel$ and $T_\perp$ are 20 to 35% lower during solar maximum as compared with solar minimum. However, the ratio $T_\perp/T_\parallel$ does not change, but the regions with a higher anisotropy are found further away from the bow shock during solar maximume conditions.

## 1.2 Earlier statistical study: Volwerk et al. (2016)

In an earlier study by Volwerk et al. (2016), MMs were studied for solar maximum conditions, using one Venus year (224 Earth days, from 1 November 2011 to 10 June 2012) and then the results were compared with the results of an earlier solar minimum study (224 Earth days, from 24 April to 31 December 2006 Volwerk et al., 2008c). As expected, the occurrence rate of MM-like[1] structures was maximum just behind the bow shock and close to the planet at the magnetic pile-up boundary. Also, it was shown that the occurrence of MM-like structures was strongly dependent on the angle between the IMF and the bow shock normal, and they mainly occur for quasi-perpendicular shocks .

By comparing the two statistically obtained results, the following conclusions were drawn about the difference between solar minimum and maximum conditions:

1. The number of MM-like structures at solar maximum is slightly higher than at solar minimum by $\sim 14\%$;

2. The observational rate for both solar conditions is the same because of the interplay of lower solar wind density and higher solar wind velocity during solar maximum than during solar minimum. One should keep in mind that cycle 24 is known to have a very weak solar maximum and thus may not be representative of more "regular" maxima;

3. The distribution of the number of MM-like structures as a function of the strength $\mathcal{B} = 2\Delta B/B$ is exponential with approximately the same coefficient for both solar conditions for "weak" MM-like structures (i.e. $\mathcal{B} \leq 1.2$). There is a less steep exponential for "strong" MM-like structures (i.e. $\mathcal{B} \geq 0.8$) with significant differences in the exponential for solar minimum and maximum;

4. Freshly created MM-like structures behind the bow shock are on average stronger for solar minimum than for solar maximum;

5. For solar minimum the general trend for MM-like structures is to decay; for solar maximum MM-like structures first grow and then decay, between the bow shock and the terminator;

6. The estimated growth rates for the MM-like structures agree well with those found for the Earth's magnetosheath.

In these past studies the MM-detection was performed with magnetometer data only. Subsequently, a coarse-grid determination of the temperature ratio $\mathcal{T} = T_\perp/T_\parallel$ by Bader et al. (2019) was used to check if the MM-like structures observational rate distribution agreed with the temperature anisotropy distribution in Venus's magnetosheath. It was shown that indeed, where $\mathcal{T}$ is large, the largest observational rates of MM-like structures was found.

In this paper we make a statistical study of the Venus Express (VEX, Svedhem et al., 2007) mission magnetometer (VEX-MAG, Zhang et al., 2006) data over the full mission from May 2006 to November 2014. Additional information is obtained through the plasma data from the ASPERA-4 instrument (Barabash et al., 2007) for both the ions and the electrons. This new

---

[1]We use the term "MM-like" throughout the paper as the usage of magnetic field only methods does not unambiguously identify MMs, whichs is only possible with plasma data at an appropriate resolution. We follow the nomenclature defined in Paper I.

and larger study is performed together with a companion paper by Simon Wedlund et al. (2023a, Paper I) which uses the same detection criteria for MM-like structures over the full MAVEN mission data at Mars, making it possible to directly compare, and for the first time, the distribution of MM-like structures at the two planets.

## 2    Detecting mirror mode-like structures in spacecraft data

### 2.1    Instrumentation

VEX was brought into a polar orbit around Venus in 2005 with an elliptical orbit and periapsis at $\sim 300$ km from the surface, which means the spacecraft entered well into the induced magnetosphere. The VEXMAG data used here have a sampling rate of 1 Hz, but are also available at 32 Hz (and for short intervals a sampling rate of 128 Hz exists). However, as the MM structures have a period of $4 \leq T_{\mathrm{mm}} \leq 15$ s (Volwerk et al., 2008b, c, 2016) the 1 Hz data downsampling is sufficient. Unfortunately, the data from the Ion Mass Analyzer (IMA) of the ASPERA-4 instrument (Barabash et al., 2007) only has a resolution of 192 s for

ions, which means that these data can only give us an indication of the overall plasma conditions (see e.g. Bader et al., 2019; Rojas Mata et al., 2022).

    In contrast, the electron spectrometer of ASPERA-4 has a cadence of 4 seconds at full energy resolution (1 eV - 20 keV) with regular switches 1-second resolution at limited energy resolution (10 - 130 eV), which would be sufficient to analyse the larger MMs for an anti-phase between magnetic field strength and electron density. Lately, Fränz et al. (2017) caculated the

electron densities for the whole Venus Express mission (where feasible), where they only used omni-directional spectra from ELS to determine the electron densities and temperatures.

    In this paper the full dataset over the VEX mission period is used, from May 2006 to November 2014, which contains both a solar minimum and solar maximum period as is shown in Fig. 1 (see also Delva et al., 2015; Volwerk et al., 2016). In this way there can be a comparison between the probability and location of MM-like structures in either solar activity period.

### 2.2    Detection method

#### 2.2.1    B-field only criteria

In order to detect the MM-like structures in the VEXMAG data we use the method introduced by Simon Wedlund et al. (2022), which is slightly different, but more accurate, from that used by Volwerk et al. (2008b). Because of the lack of high-time resolution ion data and the limited plasma (electron and ion) data availability, the detection method is based on magnetic field

measurements only. We use the same criteria as in Table 1 of Paper I, which are based on several previous works including Lucek et al. (1999a, b) and Volwerk et al. (2008b) and expanded on by Simon Wedlund et al. (2022).

    1. The magnetic field data, $\mathbf{B}$, are low-pass filtered with a two-minute-wide Butterworth filter to determine the background field $\mathbf{B}_{\mathrm{bg}}$, with $|\mathbf{B}_{\mathrm{bg}}| > 5$ nT to isolate magnetosheath conditions from average solar wind values;

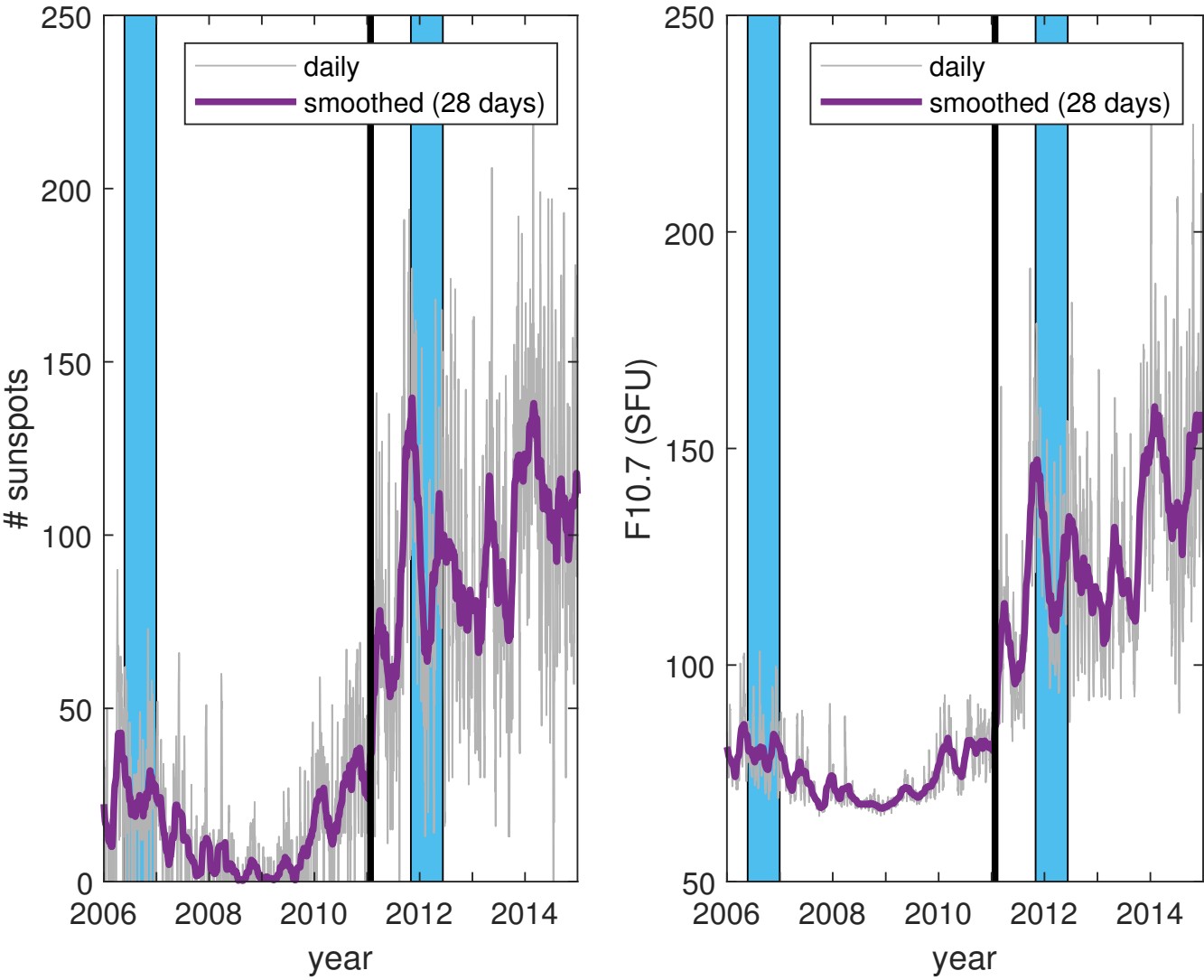

**Figure 1.** The daily (grey) sunspot numbers (left) and F10.7 flux (right) over the duration of the VEX mission (2006 - 2014) and the smoothed (over 28 days) numbers (red). The vertical line at 25 January 2011 is marking the (slightly arbitrary) boundary between the solar minimum and solar maximum periods, split at # sunspots = 50, or F10.7 = 100 SFU. The two blue boxes show the intervals discussed in Volwerk et al. (2016).

2. From the data and the background field we calculate $\Delta|\mathbf{B}|/|\mathbf{B}_{\mathrm{bg}}| = (|\mathbf{B}| - |\mathbf{B}_{\mathrm{bg}}|)/|\mathbf{B}_{\mathrm{bg}}|$, where a threshold is set to $\Delta|\mathbf{B}|/|\mathbf{B}_{\mathrm{bg}}| \geq 0.15$ (compressibility of the structure)[2];

3. Then we apply a Minimum Variance Analysis (MVA, Sonnerup and Scheible, 1998) on 15-s wide sliding windows with a 1-s shift, to obtain the directions of the minumum and maximum variations. A requirement on the maximum, minimum and intermediate eigenvalues is set to $\lambda_{\max}/\lambda_{\mathrm{int}} \geq 3$ and $\lambda_{\mathrm{int}}/\lambda_{\min} \leq 8$;

4. The angles between the minimum/maximum variation direction and the background magnetic field should be: $\Phi_{\mathrm{minV}} \geq 70°$ and $\Theta_{\mathrm{maxV}} \leq 20°$;

5. The azimuth $az = \arctan(B_y/B_z)$ and elevation $el = \arctan\left(B_z/\sqrt{B_x^2 + B_y^2}\right)$ of the magnetic field are caculated and for MM-like structures it is expected that the rotation of the magnetic field over the structure $\Delta(az, el) \leq 10°$.

The reasons of these choices above are explained in more detail in Paper I, to which the reader is referred.

### 2.2.2    Removal of false positive detections

In order to find when the spacecraft is in Venus's magnetosheath, the database of calculated bow shock crossings based on the models by Zhang et al. (2008) and Russell et al. (1988). There are more recent, more advanced, models for the bow shock, e.g., Martinecz et al. (2009) which was further developed by Chai et al. (2014). However, because of the relatively low variability of the Venus bow shock's position (Martinecz et al., 2009; Chai et al., 2014, less than $\pm 0.2 R_{\mathrm{V}}$) we expect that a simple margin on the crossing times of $\sim 30$ min, which corresponds to 14400 km, or 2.3 $R_{\mathrm{V}}$ (with $R_{\mathrm{V}} = 6051.8$ km, Venus's radius), ten times the variability, ensures that we capture the true bow shock in the data, there is no use to include these models.

We should note that around the bow shock there are other kinds of structures which have similar characteristics in polarization and compression as the MMs that we are looking for. These waves are also linked to pickup ion processes, in the case of the quasi-parallel shock. Consequently, looking at the magnetic field only, strong fluctuations ($\Delta|\mathbf{B}|/B_{\mathrm{bg}} \geq 0.1$) may appear, which are not sinusoidal and have a quasi-linear polarisation, without these structures necessarily being MMs. Most of the time, magnetic field intensity and plasma density will typically be in phase, as opposed to the expected MM anti-phase behaviour (Hasegawa, 1969). However, this information is sometimes neither available at the desired high time resolution (as in our case with VEX data) nor practical to derive as in large statistical surveys.

As in paper I, two strategies can be made in order to exclude non-MM signatures: (1) making sure that the magnetic field across the structures' region does not rotate more than 10–20°, as theoretically predicted for MMs (Treumann et al., 2004) and in agreement with past observations (Tsurutani et al., 2011), and (2) restricting the detections to magnetosheath conditions only and excluding the region around the bow shock to avoid these foreshock transients.

Strategy (1) constrains the detected structures to behaviours more reminiscent of MMs: we apply criterion 5 listed in Sect. 2.2, which ensures that the magnetic field does not rotate significantly across the structure. From the magnetic field vector the

---

[2]Note that there is a difference of a factor 2 compared to the papers by Volwerk et al.

magnetic azimuth and elevation angles are defined as:

$$az = \arctan\left(B_y/B_x\right) \qquad (4)$$

and

$$el = \arctan\left(B_z/\sqrt{B_x^2 + B_y^2}\right) \qquad (5)$$

First, we define "detection periods", which contain structures detected within a maximum of 30 s between one another and ignore isolated singular events; two separate regions are thus more than 30 s apart. This particular value of 30 s was chosen empirically as double the length of the longest MM structures found at Mars or Venus (see for example Simon Wedlund et al., 2022; Volwerk et al., 2008c, 2016); moreover this ensures that rotations could be calculated for trains of MM-like structures for which the 2-min windowed background magnetic field values would be representative. We then estimate how much azimuth and elevation angles fluctuate at the detected position of the candidate structure by calculating their running standard deviation $\langle\sigma(az,el)\rangle$ over a 2-min sliding interval, keeping only those structures where $\langle\sigma(az,el)\rangle$ is less than $10°$ for each angle (Simon Wedlund et al., 2022). This analysis of the data will be called the CSW-method.

Complementarily, strategy (2) makes use of the position of the bow shock crossing in the spacecraft data and ignores the detected structures in a range of radial distances around it (or equivalently, in a range of durations around the time of the crossing).

For Mars the automatic bow shock predictor-corrector algorithm based on magnetic field measurements only was used, explained in Simon Wedlund et al. (2022). This analys has not (yet) been done for Venus and thus this product does not exist. Therefore, only the first strategy has been applied in this paper.

### 2.2.3 Examples

Figure 2 shows a three-minute interval of VEXMAG data on 5 May 2006, where VEX is in Venus's magnetosheath (see also Volwerk et al., 2008c, Fig. 1), where the selection criteria by Volwerk et al. (2008b, grey) and by Simon Wedlund et al. (2022, green) are compared. It is clear that in the magnetosheath, around 01:17 UT, both methods find the same MM-like structures. The old criteria capture events in the shock and just behind it which are obvious false positives, while they are filtered out in the new method. Moreover, the reason we remove also the events around 01:16 UT is that the eigenvalue ratios (and thus the more stringent linear polarisation criteria) are not fulfilled any more in the new method. Because of the additional restrictions, the CSW-method identifies fewer, but more fully-formed MM-like structures.

In Fig. 3 the inbound part of a solar maximum orbit of VEX is shown, where there are some determinations of MM-like structures close to the BS and further inside the magnetosheath. Here we notice that both methods basically find the same events. There are no rejections using the CSW method.

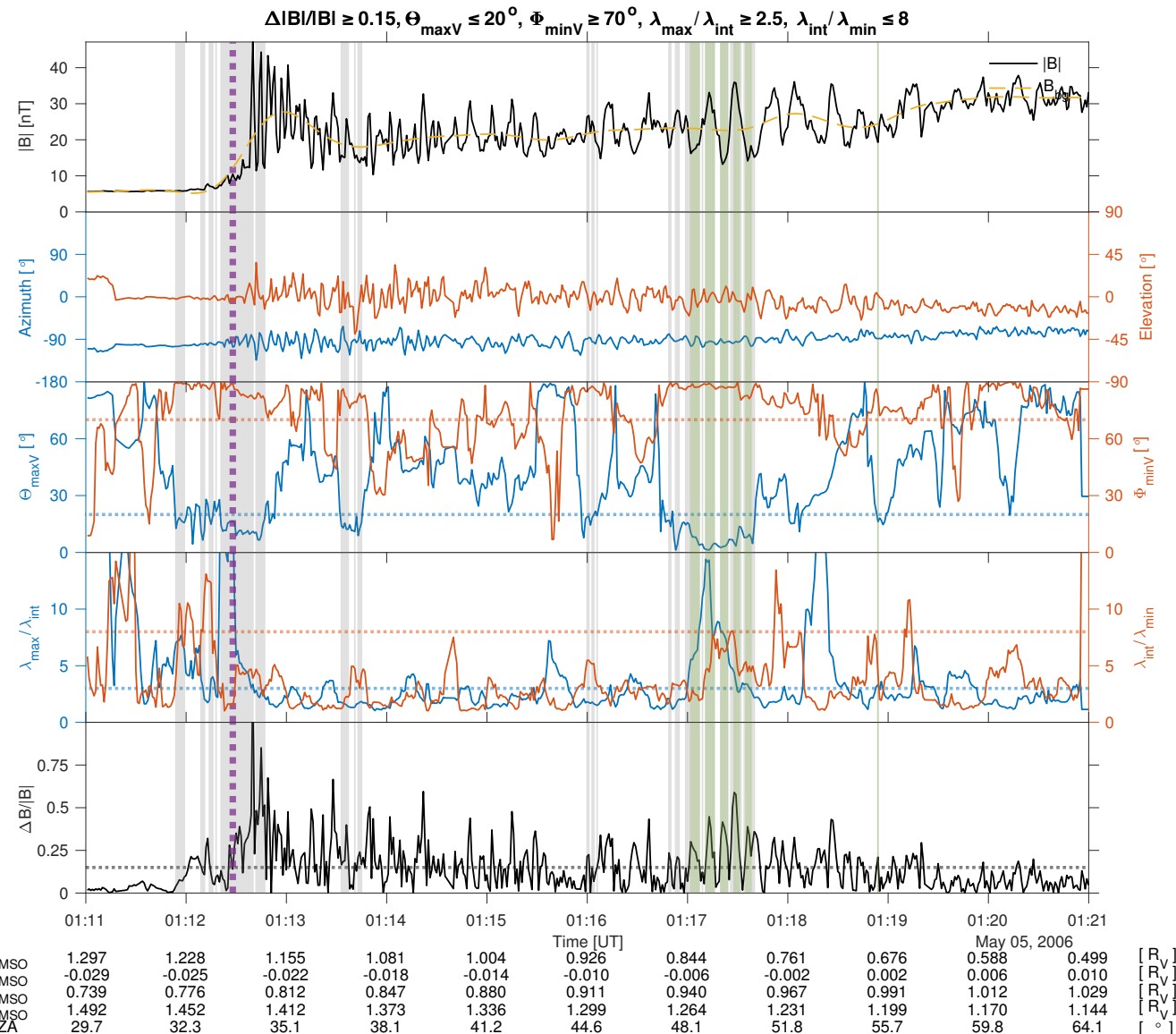

**Figure 2.** Ten minutes on 5 May 2006 when VEX entered through the bow shock into the magnetosheath. Shown are (a) The total magentic field; (b) The azimuth and elevation of the magnetic field; (c) The angles of the minimum and maximum variation direction with the background magnetic field; (d) the ratios of the eigen values and (e) the $\Delta B/B$. The grey shadings show the events found with the criteria in Volwerk et al. (2008b) and the green shadings show those found with the criteria in Simon Wedlund et al. (2022) (with the green shading overlapping the grey shading). The purple vertical dotted line is the model-location of the nominal bow shock. Note that for this event there are no electron data available.

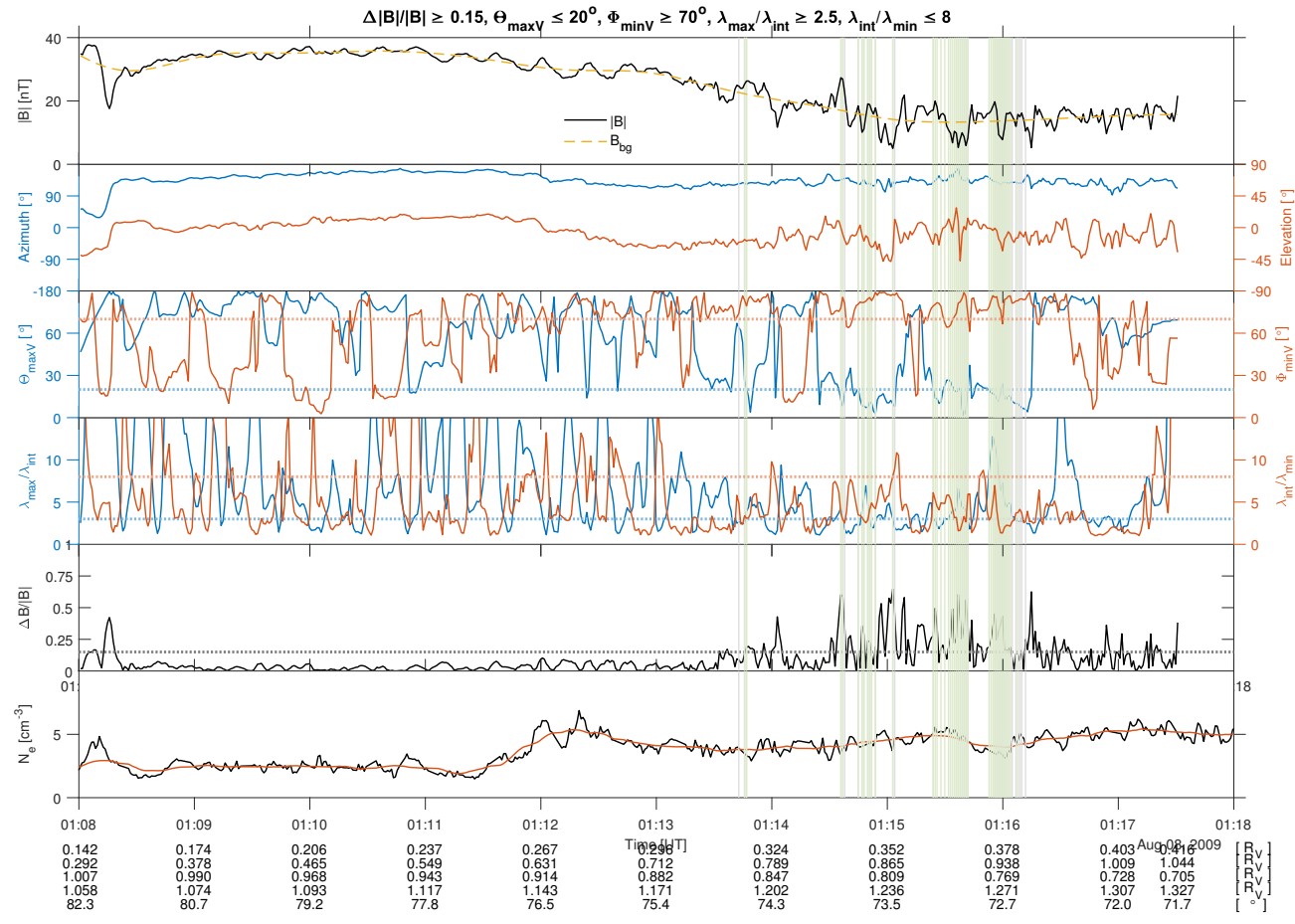

**Figure 3.** Ten minutes on 8 August 2009 when VEX returned from pericentre through the magnetic pile-up region close to Venus. Same format as Fig. 2. For this event there are electron data from ASPERA-4 ELS in the bottom panel.

**Table 1.** Total number of MM-like structures in the VEXMAG dataset (equivalent to a duration in s because of the magnetometer resolution of 1 s) and residence times in the magnetosheath, probability $\mathcal{P}$ of observing mirror mode structures during that time, and averaged MM depth $\langle \Delta B/B \rangle$ for solar minimum and maximum conditions.

| period | Total MM-like structures # | $\Delta T^{\text{sc}}$ seconds days | Probability % | Average $\langle \Delta B/B \rangle$ |
|---|---|---|---|---|
| Sol. Min. | 67,250 | 8,441,391 97.7 | $\sim 0.8$ | $0.14 \pm 0.12$ |
| Sol. Max | 74,739 | 7,611,965 88.1 | $\sim 1.0$ | $0.18 \pm 0.20$ |

## 2.3 Mapping technique

Following the mapping technique of Paper I, the results are shown on a grid in cylindrical coordinates based on the VSO coordinate system[3], i.e. $X_{\text{VSO}}$ and $R_{\text{VSO}} = \sqrt{Y_{\text{VSO}}^2 + Z_{\text{VSO}}^2}$ with a size of $0.1 \times 0.1\,R_{\text{V}}$. For each grid cell the total number of seconds for which the MM criteria are fulfilled, $\Delta T^{\text{struct}}$, is calculated as well as the total residence time of VEX in that box, $\Delta T^{\text{sc}}$. Both determinations are done for solar minimum and solar maximum. The probability of MM-like structures is then simply calculated from the ratio of the two:

$$\mathcal{P} = \frac{\Delta T^{\text{struct}}}{\Delta T^{\text{sc}}}. \tag{6}$$

We only consider grid cells where the spacecraft stayed at least 30 minutes in cumulated time, to ensure good statistics throughout.

A first quick result can be obtained by looking at the total duration of MM-like structures spanning years 2004-2016 of VEXMAG data. as shown in Table 1. The total residence time shows that VEX stayed longer in the magnetosheath at solar minimum than at solar maximum. This is caused by the asymmetric division between solar miminum and maximum of the VEX mission (see Fig. 1) and is influenced by a change in attitude of the orbit over the duration of the mission, where the semi-major axis slowly rotated further southward and in the late stage of the mission back northward again. Nevertheless, more events are found and the total observational rate is $\sim 50\%$ higher during solar maximum. Note that, although these are both multiple years of orbits, the time spent in the magnetosheath is actually rather short, 185.8 days out of 3195, because of the highly elliptical orbit with a periapsis and apoapsis of 250 km ($1.04\,R_{\text{V}}$) and 66000 km ($11.90\,R_{\text{V}}$), respectively, whereas the magnetosheath spans typical radial distances of 1.1 to 1.5-3 $R_{\text{V}}$.

---

[3]VSO: Venus-Solar-Orbital coordinate system, where $X_{\text{VSO}}$ points towards the Sun, $Z_{\text{VSO}}$ is towards solar north and $Y_{\text{VSO}}$ completes the triad and is in the opposite direction of Venus's orbit.

In Figs. 7 and 8 (left panels) the total residence time of the spacecraft in $0.1 \times 0.1\,R_V$ cells is shown for solar minimum and maximum. It shows that there is a slight difference of total residence times in e.g. the polar region (at $X_{\mathrm{VSO}} = 0$), where VEX spent relatively less time during the solar maximum interval.

### 2.4 Calculating MM-like observational probability

The overall numbers and probabilities only give a small indication that MM-like structures are more prone to be excited during solar maximum as compared to solar minimum. The interesting part is when the probability per $0.1 \times 0.1\,R_V$ cell is examined.

Figures 7 and 8 show the statistical results of our search for solar minimum and maximum respectively. The left panels show the total residence time $\Delta T^{\mathrm{sc}}$ of VEX in each grid cell. The middle panel shows the probability of MM-like structures per cell calculated with Eq. (6). The right panel shows the mean $\langle \Delta B/B \rangle$ in each grid cell, limited by the restriction that $\langle \Delta B/B_{\mathrm{bg}} \rangle \geq 0.15$.

There are clearly two regions on the dayside where the MM-like structures are most prevalent: for solar minimum right behind the BS and close to the ionopause, and for solar maximum also behind the bow shock, but slightly deeper inside the magnetosheath, and again at the magnetopause/ionopause. In a marginal way, there is also a third area behind the planet, around $(X, R) = (-2, 0)R_V$, where MM-like structures seem to be present for solar maximum, which is not so prominent at solar minimum. One may assume, in a first approximation, that the structures are observed where they are generated, that is, that the creation of the first two regions is caused by two anisotropic energization mechanisms of the ions: close to the bow shock the perpendicular temperature is enhanced by preferential heating along the perpendicular direction to the magnetic field of ions crossing the quasi-perpendicular BS; close to the magnetopause/ionopause the perpendicular temperature is enhanced by the magnetic pile-up in front of the planet and the conservation of the first adiabatic invariant.

### 2.5 Controlling parameters

The presence of MM-like structures in Venus's magnetosheath is first of all dependent on the type of bow shock. A quasi-perpendicular bow shock has its normal nearly perpendicular to the impinging IMF. In this case the picked-up protons in the solar wind are energized mainly in the direction perpendicular to the magnetic field. This increases the $T_\perp/T_\parallel$ term in the instability criterion, Eq. (1), and thus MM-like structures are expected to be generated. However, this criterion is not a sufficient condition, as was shown in the data from the Solar Orbiter flyby of Venus, where behind a near-perpendicular bow shock, ion cyclotron waves (ICW) were generated (Volwerk et al., 2021) instead of MM-like structures. This was caused by a low plasma-$\beta(\approx 1.3)$ behind the bow shock, in agreement with Gary et al. (1993), who showed that, for low plasma-$\beta$, the ratio $T_\perp/T_\parallel$ must be $\sim 15\%$ larger for MM generation than for ICW generation.

Behind a quasi-parallel bow shock the generation of MM-like structures is not expected to be significant due to the lack of perpendicular energization of the protons, which was shown by Volwerk et al. (2008c). In this condition, pickup ion effects alone may lead to temperature anisotropies able to generate MMs (Gary, 1992).

Venus's orbit has an excentricity of $\epsilon \approx 0.0068$, which means that, unlike Mars with an eccentricity, $\epsilon \approx 0.0934$, seasonal effects are not expected. However, the average bow shock location for solar minimum and maximum is significantly different.

For example, the terminator distance is $R_{\rm t,min} \approx 2.14\,R_{\rm V}$ (Zhang et al., 2008) and $R_{\rm t,max} \approx 2.40\,R_{\rm V}$ (Russell et al., 1988):
for solar maximum conditions, the bow shock significantly expands in the solar wind and inflates by more than 10%. The difference between the two solar activities is clearly seen through the further distance into the magnetosheath of the MM probability peak for solar maximum.

It should be noted, however, that the maximum of solar cycle 24 was (much) weaker then previous solar maxima (see e.g., McComas et al., 2013), which means that the solar wind conditions may not be representative of a "regular" solar maximum.

## 3 Results

### 3.1 Overview of the full dataset

In Fig. 4 we show the Probability Distribution Function (PDF) of all variables necessary in the selection criteria listed in Sect. 2.2, for the whole mission. The maxima of the PDFs have been indicated by a grey bar in the panels. These PDFs need to be checked against the selection criteria. Panels (b) through (f) of the histograms show only that the bulk of MM-like events peaks at around 7 nT of B-field (nominal magnetosheath values), at $\Delta B/B$ of 0.17 (threshold being 0.15), with a perpendicular direction to the minimum variance direction and a very broad, flat distribution of maximum variance directions between $10°$ and $20°$ (peak at $16°$).

Putting indeed together all criteria, we obtain the official number of MM-like structures in Venus's magnetosheath. In Fig. 5 we show the daily occurrence rate of these structures over the whole VEX mission, with overplotted in red a 7-day running average. The average number of observed events per day is $\langle N \rangle = 26 \pm 23$. However, as mentioned earlier, VEX only spends 185.8 days out of 3195 in the magnetosheath, i.e. $\sim 6\%$ of the spacecraft orbiting time. Assuming on average a mirror mode structure to last 10 s, we end up with about 26 mirror mode structures observable per day. This means that to obtain the total number of events per day, we have to correct this by multiplying this number by a factor 100/6, which leads to $\langle N_{\rm corr} \rangle \approx 430$ 1-second events per day.

Figures 6, 7 and 8 show the overall results of our analysis, for the whole data set, and for solar minimum and maximum conditions, respectively. Displayed are the residence time of VEX around Venus, the probability $\mathcal{P}$ to find MM-like structures in the $X - R$-plane and the average depth of $\Delta B/B$ in each grid cell. Figure 9 displays the absolute difference of detection probability between high solar activity and low solar activity.

The probability of MM-like structures for solar minimum and maximum, respectively, for the cleaned data set, i.e. with the requirements 1 – 5 from Sect. 2.2 applied is calculated. As mentioned earlier, there seem to be two regions in which this rate is greatly enhanced compared to the rest of Venus's surroundings: just behind the bow shock and around the ionopause. Similarly, Figs. 7 and 8 (right panels) show the average depth of the observed MM-like structures in each bin. As shown in Table 1 the average values for $\langle \Delta B/B \rangle$ are rather small for this data set, however the full distribution of $\Delta B/B$ is shown in Fig. 10.

First, we look at the depth of the MM-like structures. Fig. 10 shows the distribution of the depth $\Delta B/B$ of the MM-like structures, where the majority of the MM-like structures falls into the greyed-out category $0.05 \leq \Delta B/B \leq 0.15$, which are not taken into account in the analysis as per the selection criteria.

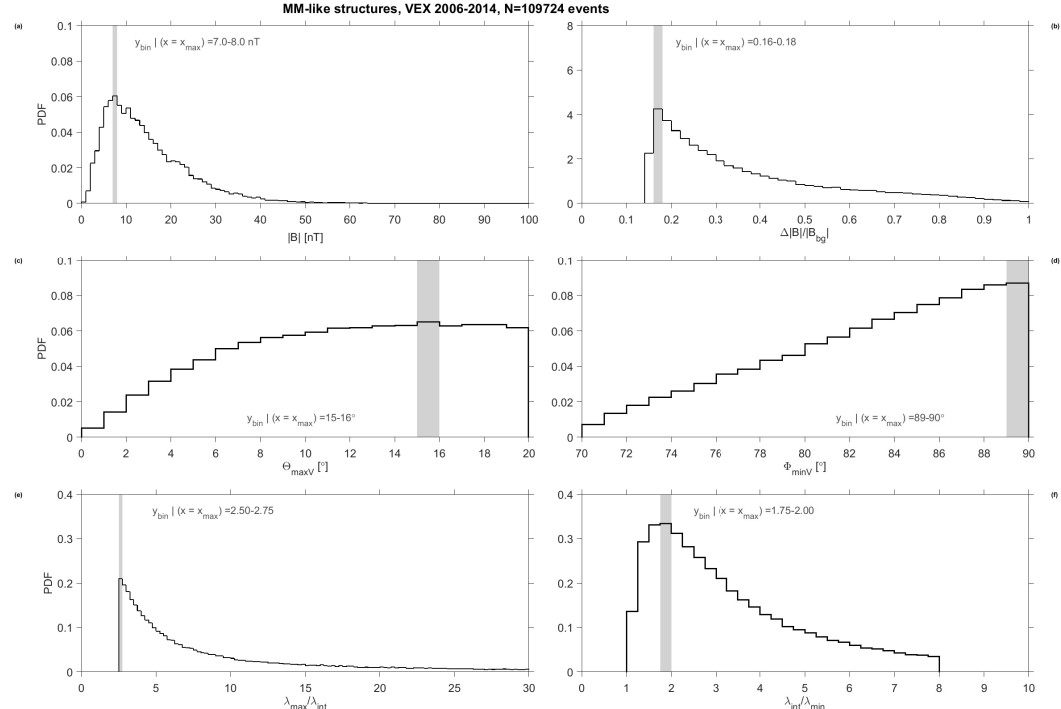

**Figure 4.** Probability Distribution Functions (PDFs) of selection criteria for MM-like structures in the VEX magnetometer data for the whole mission. (a): Total magnetic field intensity $|\mathbf{B}|$, in bins of 1 nT. (b) Magnetic field fluctuations $\Delta|\mathbf{B}|/B_{\mathrm{bg}}$, in bins of 0.02. (c) and (d): Angles between average magnetic field direction and maximum (minimum) variance direction $\Theta_{\mathrm{maxV}}$ ($\Phi_{\mathrm{minV}}$), in bins of $1°$. (e) and (f): Ratios of maximum to intermediate $\lambda_{\mathrm{max}}/\lambda_{\mathrm{int}}$ (intermediate to minimum, $\lambda_{\mathrm{int}}/\lambda_{\mathrm{min}}$) eigenvalues, in bins of $0.25$. The position of the maximum of the PDF and its typical bin is marked by a grey zone. All bins are uniformly distributed.

Both distributions are very similar percentage-wise, indicating that solar activity has little influence on the depth. However, one can discern a dichotomy in the percentages between solar minimum and maximum. The green bars, describing the ratio of the percentages of solar minimum and maximum ($\mathcal{G}$ = blue/red, multiplied by 10 here for visibility) show that up until

$\Delta B/B = 0.5$, there is a higher percentage for solar minimum, $\mathcal{G} > 1$, and after that for solar maximum, $\mathcal{G} < 1$. It was shown above that the location of the MM-like structures is different as well as the total amount of MM-like structures measured: 38,901 and 64,883, respectively (see Table 1). Especially, very deep events, $\Delta B/B \geq 0.75$, are $\sim 5$ times more abundant at solar maximum as compared to solar minimum, with 1973 and 357 events, respectively.

## 3.2 Dependence on solar activity

It was shown above, in Figs. 6 - 9, that the main differences of the probability of MM-like structures for solar minimum and maximum are: (1) the total number of events measured, and (2) the location within the magnetosheath where they were

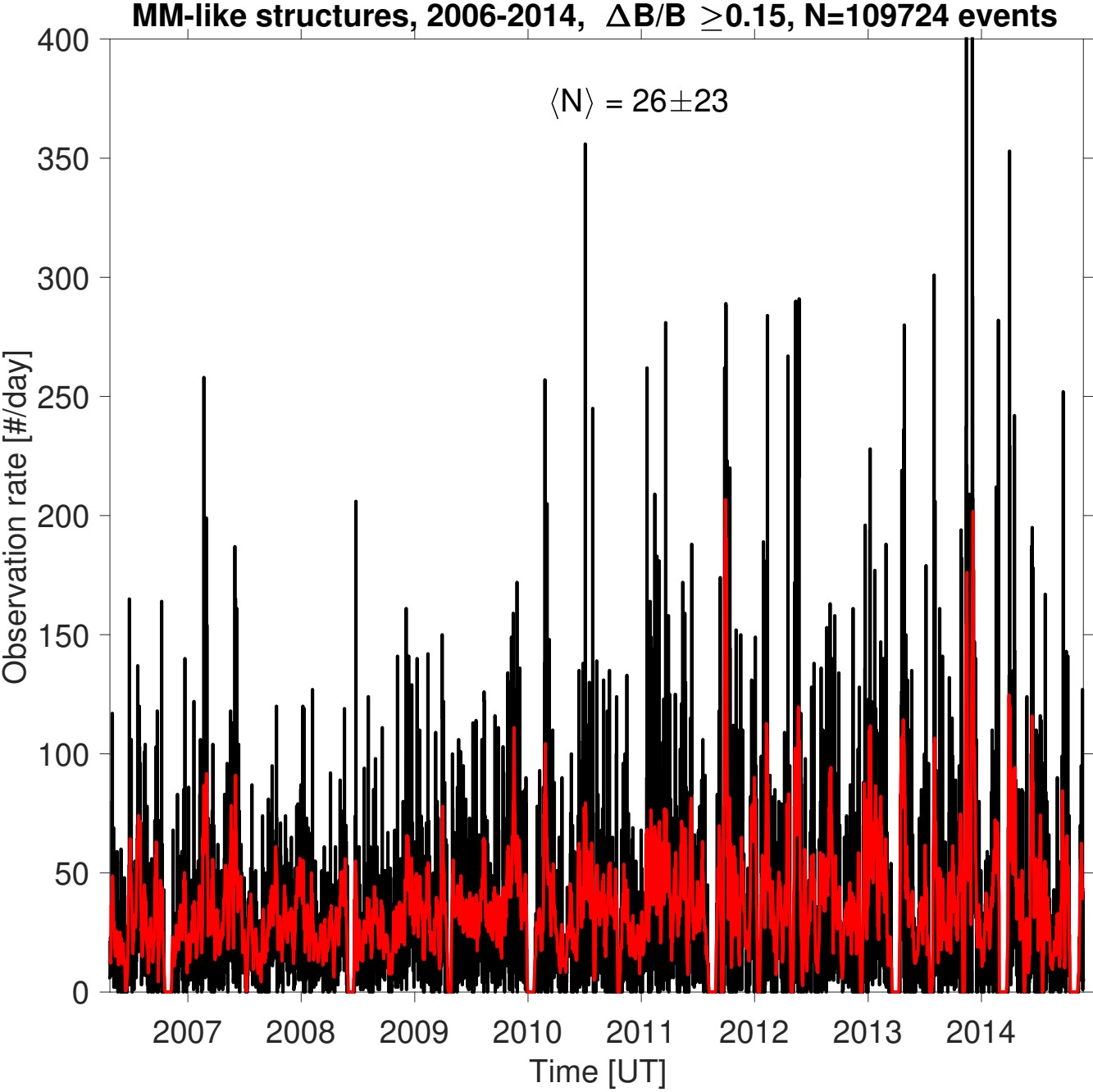

**Figure 5.** Daily observation rate of MM-like structures as observed by the VEX magnetometer for the whole mission, using the selection criteria mentioned in Sect. 2.2. The red line corresponds to the running mean of the black curve over 7 days. $\langle N \rangle$ is the median of the signal in black, with its corresponding standard deviation.

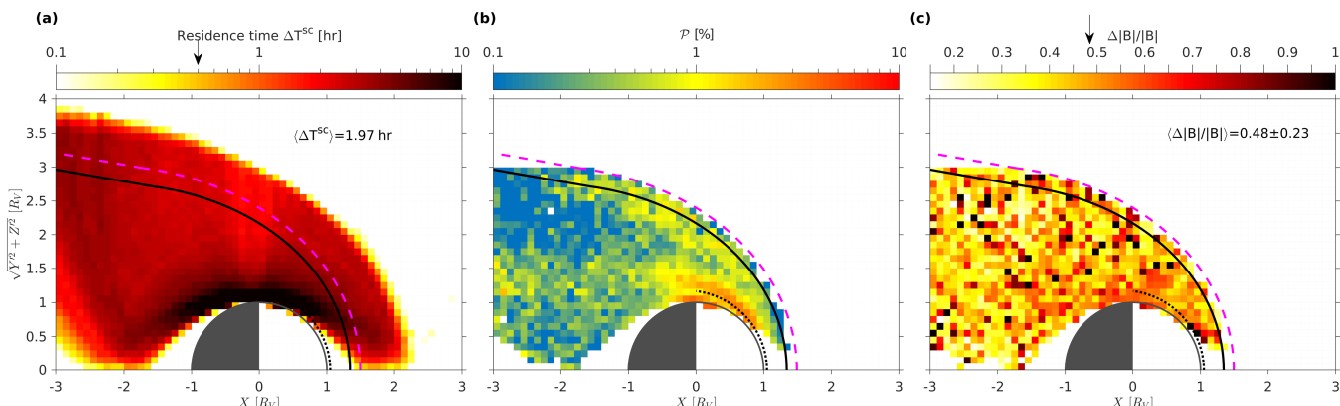

**Figure 6. Full data set**: (left) The total residence time of Venus Express in the $X - R$-plane. The thick black line is the bow shock location as determined by Zhang et al. (2008) for solar minimum. The dotted line is the magnetopause/ionopause location.(middle). The dashed magenta line shows the location of the solar maximum bow shock. The probability of MM-like structures in the $X - R$-plane. There are two clear regions of increased $\mathcal{P}$, just behind the bow shock and close to the magnetopause/ionopause. (right) The average depth of the mirror modes in each grid cell, limited by $\Delta B/B \geq 0.15$.

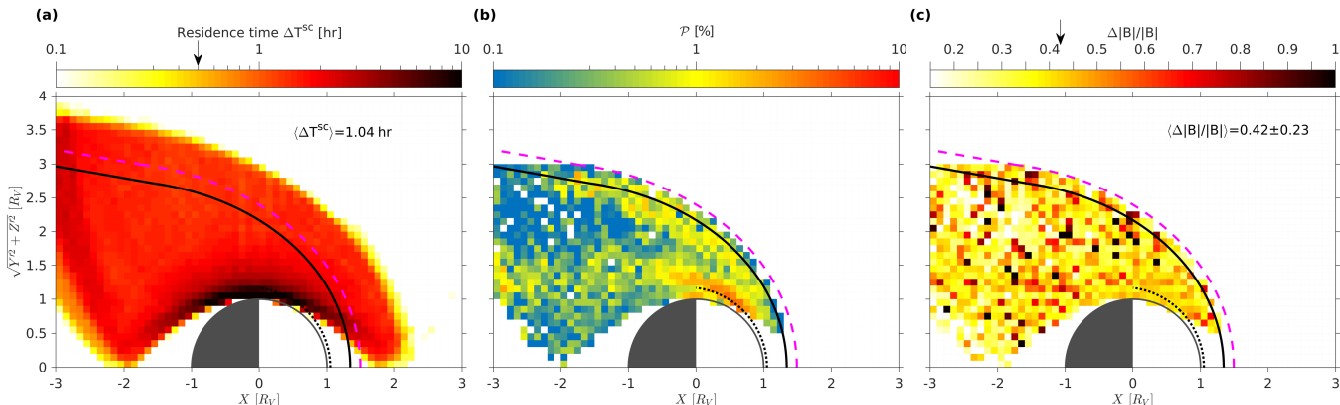

**Figure 7. Solar Minimum**: same format as Fig. 6.

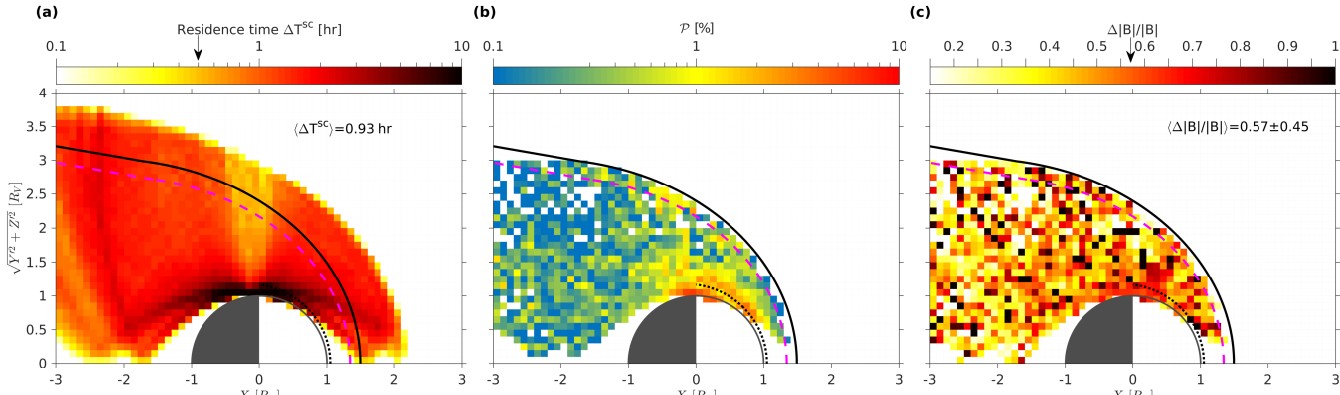

**Figure 8. Solar Maximum**: same format as Fig. 6.

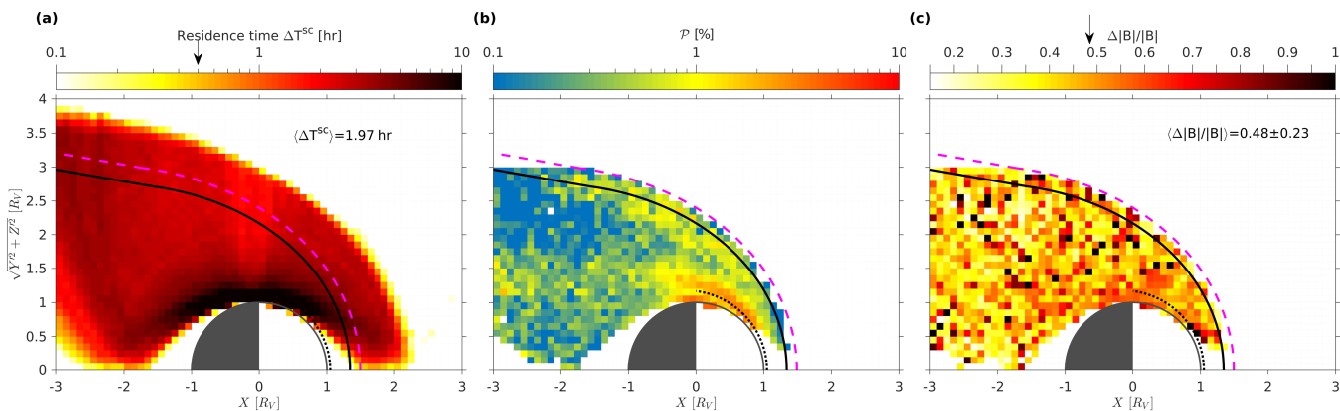

**Figure 9.** (a) The total residence time of VEX in hours for the whole dataset. (b) The difference between solar maximum and minimum in probabilities, calculated through $(\mathcal{P}_{\mathrm{hi}} - \mathcal{P}_{\mathrm{lo}})/(\mathcal{P}_{\mathrm{hi}} + \mathcal{P}_{\mathrm{lo}})$, with the red hues showing where the solar maximum conditions are dominating in that box, and the blue hues where solar minimum conditions are dominant. (c) The $\Delta B/B$ for the whole dataset.

observed. As previously pointed out, no seasonal variations in the probability is expected. However, there can be other effects that can have an influence on the probability of MM-like structures.

As the MM-like structures behind the BS are mainly generated by freshly created pick-up ions, re-energized by their crossing
of the bow shock, it stands to reason that the solar EUV flux plays a role, as photo-ionization is the main source for these particles. Indeed, it was shown by Delva et al. (2015) that the higher number of observed proton cyclotron waves for solar maximum, as compared to minimum, was caused by the higher EUV flux, supplying a greater number of newborn protons from Venus's exosphere.

The split between solar minimum and maximum based on sunspots is rather arbitrary and a more sophisticated method can be
used to study the influence of solar activity, through the daily F10.7 flux. As can be seen in Fig. 1, however, the divide assumed in this paper splits the periods well with #sunspots $\lesssim 50$, or $F10.7 \lesssim 100$ SFU. Every event is assigned its corresponding

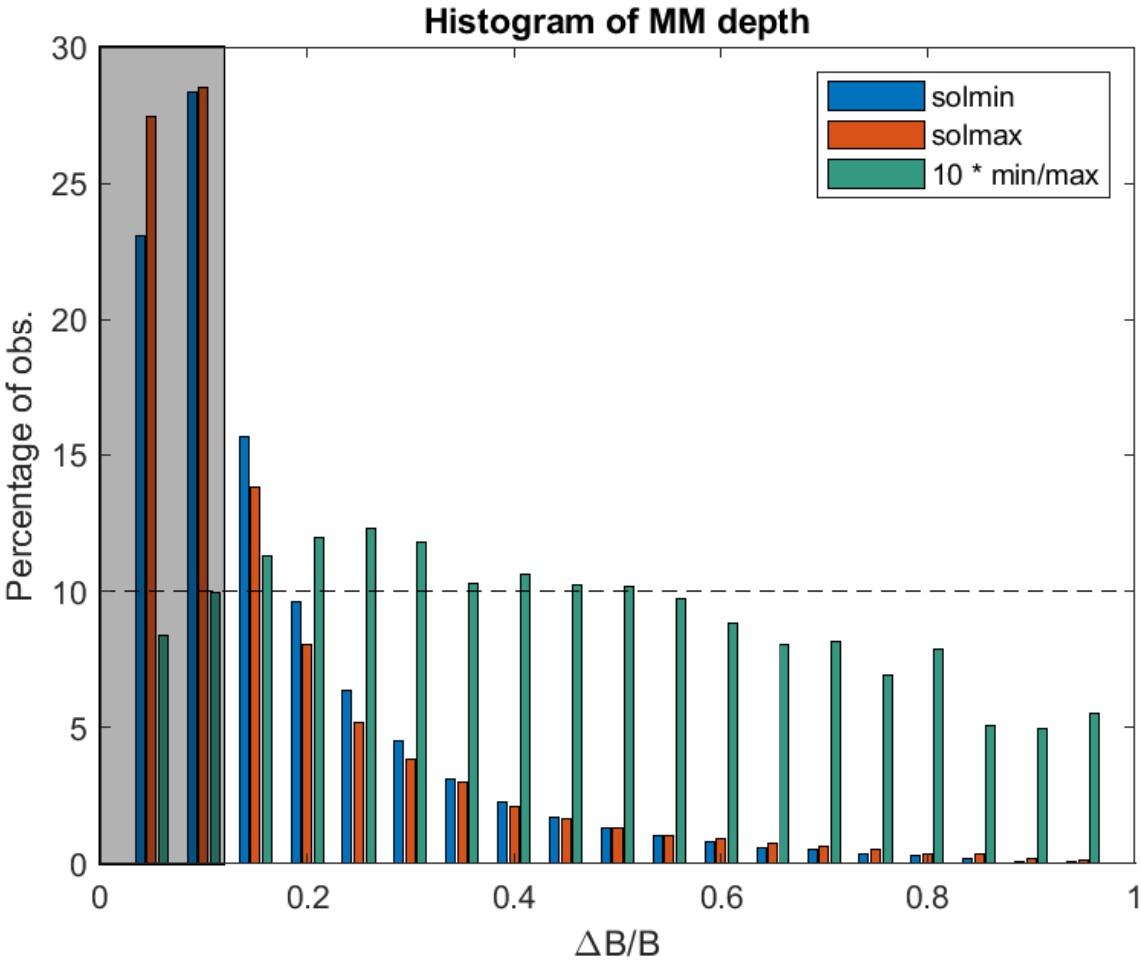

**Figure 10.** Histograms of the distribution of the depth $\Delta B/B$ of the MM-like structures for solar minimum (blue) and maximum (red) and the ratio between the two (green = blue/red, multiplied by 10 for visibility). The grey shaded region shows the structures with $\Delta B/B < 0.15$ that are not taken into account in the analysis in this paper.

daily averaged F10.7 value. Figure 11 shows the histogram of the daily F10.7 value over the whole mission in percentages of the total days, as well as that for the events in percentages of the total number of events. It is clear that both histograms show a similar trend, with only slight difference of a few percentage-points. The yellow bars show the ratio of percentage event flux over percentage daily flux $\mathcal{Y}$ (yellow = red/blue, multiplied by 10 for visibility). The dotted vertical line shows the division between solar minimum and maximum. The average ratio for solar minimum is $\overline{\mathcal{Y}} \approx 0.9$, whereas for solar maximum $\overline{\mathcal{Y}} \approx 1.5$ (limiting to values $F10.7 \leq 200$ SFU). This means that there is a slight influence of the F10.7 flux onto the creation of MM-like structures, such that at solar maximum the structures are prone to exist for higher flux, whereas for solar minimum both the daily and event fluxes are basically equal.

## 3.3  Dependence on bow shock characterisation

In the introduction we stated that MMs mainly occur during periods of quasi-perpendicular bow shock conditions, as was shown in the Earth's magnetosheath by Génot et al. (2008) and at Venus by Volwerk et al. (2008c). Without a solar wind monitor upstream of Venus, it is not possible to obtain the simultaneous IMF for each event. Therefore, in this study, we have determined the IMF before the inbound and after the outbound bow shock crossing. With these upstream magnetic fields the character of the bow shock can be obtained: quasi-perpendicular or -parallel. Then, for each MM event the nearest-in-time bow shock crossing is sought to characterise under which conditions the MMs are created (for the bow shock database, see Simon Wedlund et al., 2023b).

In the overall 109,724 events for which the IMF could be determined it was found that 81,272, or $\sim 80\%$, are linked to a quasi-perpendicular bow shock. How this influences the observational rate of MMs is shown in Fig. 12. Here the events are split into three groups: quasi-perpendicular with $30°$ around the perpendicular direction to the normal, quasi-parallel with $30°$ around the normal direction, and intermediate for the remaining $30°$ wide bins.

Looking at the number of events in these three categories, we find that quasi-perpendicular conditions contains $\sim 29\%$, intermediate conditions $\sim 68\%$ and quasi-parallel conditions $\sim 3\%$ of recorden MM-like events. The observational rate of the MMs is here calculated by dividing the number of events by the reduced residence times of VEX, based on the percentage of events found. As expected there is a very strong reduction in the observational rate for the quasi-parallel bow shock. There is also a reduction of MM-like events for the quasi-perpendicular bow shock. Interesting is the much higher occurrence of events in the intermediate category.

## 4  Electron density data

One of the characteristics of MMs is that there is an anti-phase between the magnetic field strength and the plasma density (see e.g., Tsurutani et al., 1982). The magnetic field only CSW-method to find MMs (as first proposed by Lucek et al., 1999a) needs to be tested for possible misinterpretations when plasma data are available. Rae et al. (2007) performed a study on the robustness of the method by Lucek et al. (1999a) and found that the **B**-field-only method, indeed, worked well. As mentioned

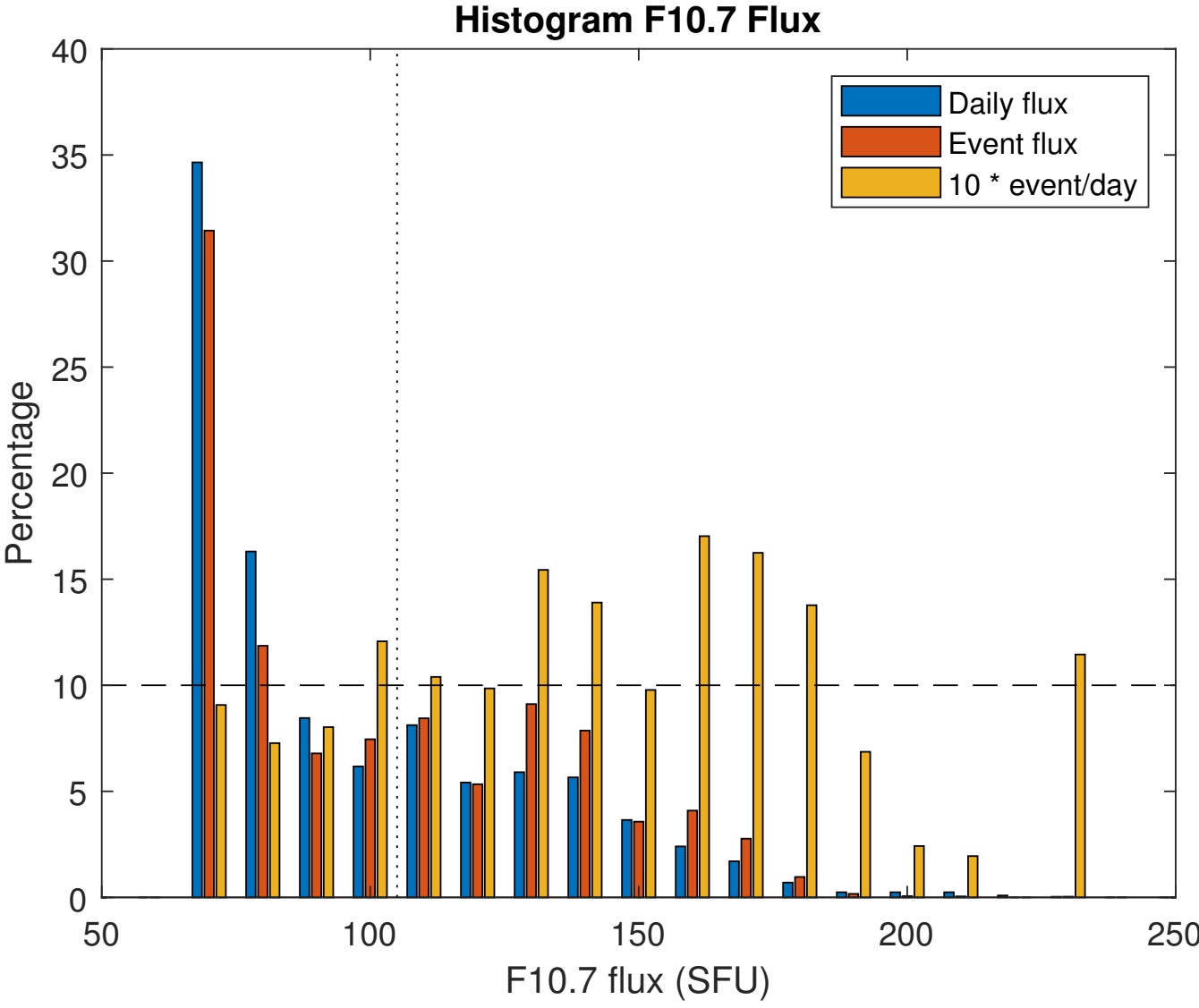

**Figure 11.** Histograms of the F10.7 flux. Combined are the distribution of the percentages of daily flux (blue) and of the flux for each event specifically (red). The yellow bars show the ratio of the two fluxes (yellow = red/blue, multiplied by 10 for visibility).

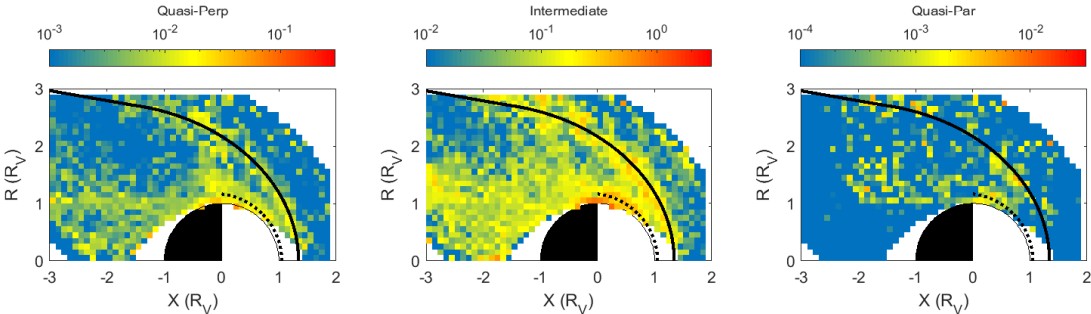

**Figure 12.** MM observational rate for quasi-perpendiculare bow shock for reduced angle bins ($\sim 30°$). Note that for visibility, the color bars have different limits in each panel.

above, Fränz et al. (2017) caculated the electron densities for the whole Venus Express mission, which we will now use to check the validity of our method.

Unfortunately, for the event on 5 May 2005 (Fig. 2) there are no electron data available, but for the near-Venus event there is (Fig. 3). In order to show the anti-phase between the magnetic field and the electron density we have resampled the magnetic field data to the same time resolution as the electron data. In order to avoid possible off-sets, we have then calculated $\Delta N/N$, which is then compared to $\Delta B/B$ in Fig. 13.

It is clear that the $\Delta N/N$ and $\Delta B/B$ are in anti-phase over most of this time interval in which the CSW-method determined the presence of MMs. Much of this interval was not selected by the CSW-method because of the strong selection criteria. But we can see that there are regions with no anti-phase, so no MM-candidates. However, there are also regions where there is an anti-phase where other strong criteria are not fulfilled, implying that we likely underestimate the total number of structures detected throughout the mission (see discussion in Paper I for the Mars case). Moreover, only parts of the full sinusoidal-like structure are usually captured by our algorithm, which further confirms this overall underestimation.

In order to check how the CSW-method compares to the wave selection used by Fränz et al. (2017), we have also analysed 3 June 2006 (Fig. 14. Fränz et al. (2017) used the wave identification method proposed by Song et al. (1994) (also used, e.g., by Ruhunusiri et al., 2015), based on the calculation of compressional and transverse wave power, plasma and magnetic pressure, and velocity variations. They find an interval of 6 min around 01:29 UT, in which there is MM activity. The green and grey lines in Fig. 14 show the identification of MMs by the CSW-method and the old conditions (Volwerk et al., 2008a). There is an anti-phase between $\Delta B/B$ and $\Delta N/N$ at the marked locations, as well as at other non-marked locations, showing the presence of MMs. This further gives confidence in our **B**-field-only detection algorithm capturing structure candidates that are indeed MMs.

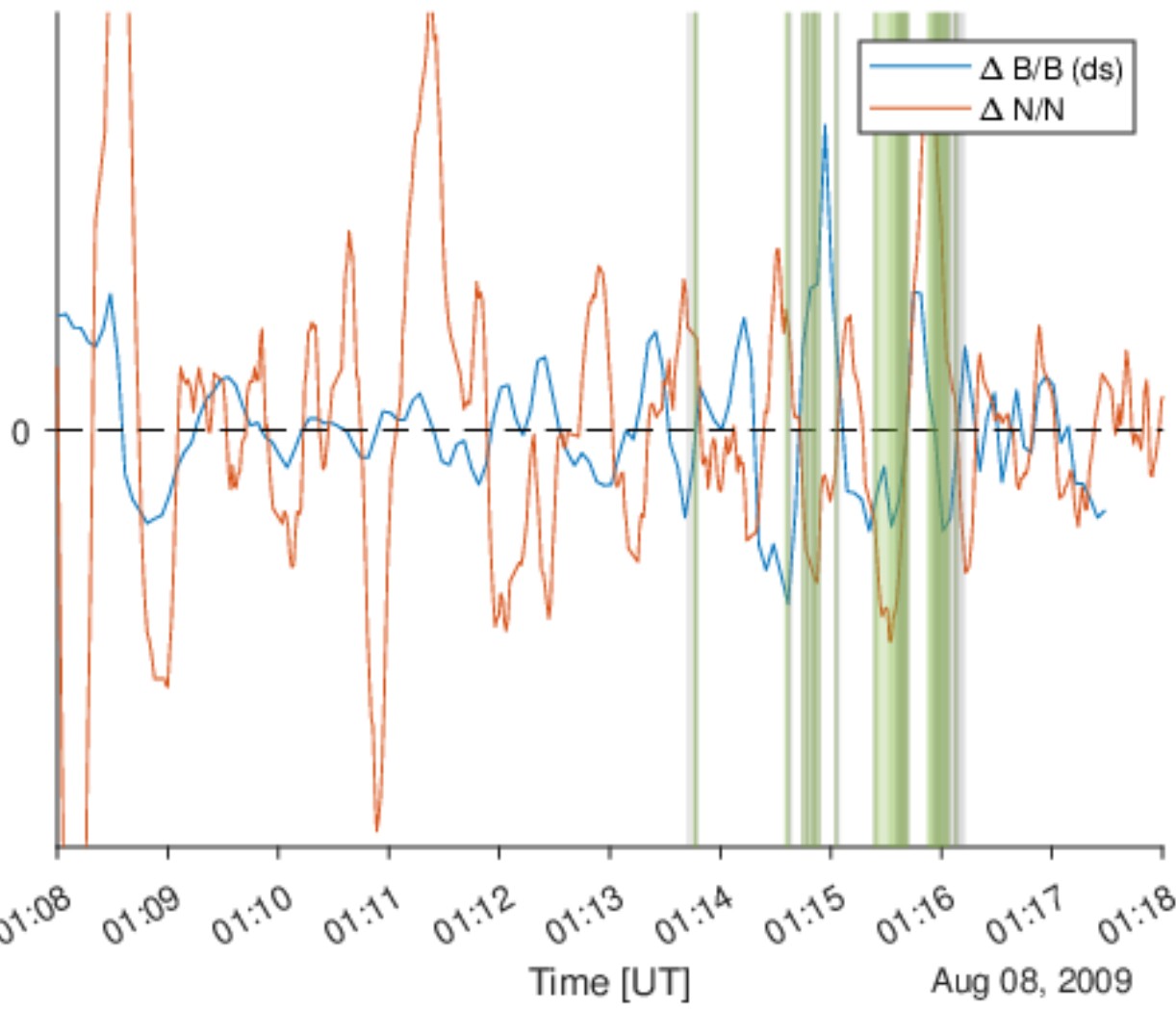

**Figure 13.** The magnetic field and electron fluctuations, $\Delta B/B$ (blue) and $\Delta N/N$ (red). The MM-active interval of Fig. 3. The anti-phase between magnetic field strength variations and electron density variations is clearly visible around the green and grey identification bars.

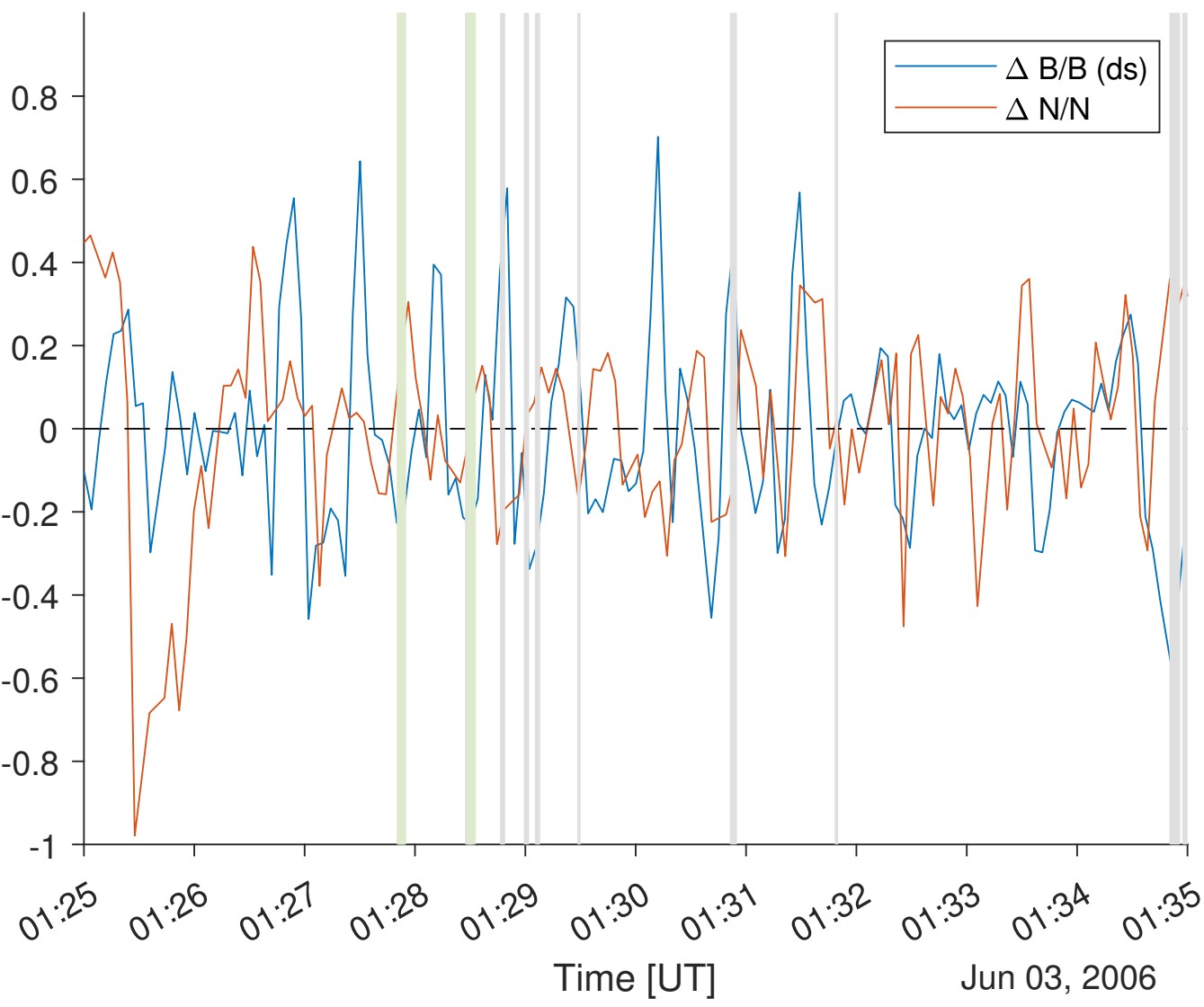

**Figure 14.** The MM-active interval shown by Fränz et al. (2017) in their Figure 1 analysed with the CSW-method.

## 5   Discussion

We have studied the probability of mirror mode like structures in Venus's magnetosheath over the whole Venus Express mission, with the strict constraints as presented in Paper I. The outcome can be compared to the previous study on MM-like structures by Volwerk et al. (2016) and a recent study of the plasma properties around Venus by Rojas Mata et al. (2022).

In this paper, and its companion (Simon Wedlund et al., 2023a) (Paper I), we have extended the "magnetometer-only" identification method of Lucek et al. (1999a) to find the MM candidates at Mars and Venus, with mitigation strategies trying to overcome the detection of waves that may behave like MMs but in effect are not (removal of false positives in Sect. 2.2.2), called the CSW-method. It is clear from this paper and Paper I that this is not a foolproof method and that the use of available plasma data increases significantly the accuracy of this method. Naturally, there are also other methods to determine the wave modes in magnetoplasma data, e.g., the one by Song et al. (1994), where the various ratios of magnetic field components (compressional and transverse) and plasma (pressure and velocity) are used. This method leans heavily on the knowledge of the ion-plasma data, which in the case of VEX is only available at an unsuitable resolution. The method itself has been criticised in several papers: Schwartz et al. (1996) points out that the step-wise down-selection of the wave mode is rather sensitive to an incorrect choice in the decision tree and advised the more involved analysis presented by Denton et al. (1995); and Denton et al. (1998) further critiques the Song et al. (1994) method.

The whole Venus Express mission extended over most of a solar cycle, where both solar minimum and maximum are sampled well, as shown in Fig. 1 and in Table 1 for $\Delta T_{\text{tot}}^{\text{sc}}$. There are two main differences between these two periods with respect to the MM-like structures: (1) The total number of events and probability are larger for solar maximum; (2) the location of observations are different with the MM-like events found deeper in the magnetosheath for solar maximum.

There is a slight difference in the distribution of the depth, $\Delta B/B$, of the MM-like structures for solar minimum and maximum (see Fig. 10). For solar minimum there are more weaker MM-like structures, whereas for solar maximum there are more stronger structures. Similarly, there is a slight dependence of the observation of MM-like structures with respect to the F10.7 flux. The event flux is higher than the daily average flux during solar maximum, whereas for solar minimum they are almost equal.

This means that only the generation of MM-like structures is strongly dependent on solar activity: more activity leads to more ionization, which in its turn leads to more ion pick-up and crossings of the instability threshold, Eq. (1). But there seems to be no evolutionary development of the MM-like structures with respect to their depth, not through increased solar activity. There could be a temporal development while they are transported by the plasma flow, which will be discussed further below in Sect. 5.1.

Looking at the locations of the maxima of the probabilities $\mathcal{P}$ in Figs. 7 and 8 one finds that the MM-like structures identified just behind the bow shock are deeper inside the magnetosheath for solar maximum than for solar minimum. In Fig. 8, middle panel, the bow shock location for solar minimum has also been indicated by a dashed magenta line. This panel shows that the maximum probability is at the location of this magenta line. It is unclear whether this is just by chance or if this location has a physical meaning.

## 5.1 Comparison with Volwerk et al. (2016)

Mirror modes in Venus's magnetosheath were first discovered by Volwerk et al. (2008b, c) and a comparison between solar minimum and solar maximum was presented in Volwerk et al. (2016). These studies, however, were based on only one Venus-year (223 Earth days) of data for each solar activity level. Nevertheless, some of the results from those papers are in agreement with the results presented above for the full 2006-2014 VEX dataset.

Figure 3 in Volwerk et al. (2016) shows the occurrence rate of the MM-like structures on a coarser grid of $0.25 \times 0.25 \, R_{\mathrm{V}}$. Note that the definition of the occurrence rate in the papers by Volwerk et al. (2016) is different than in our present study. They gathered together closely spaced intervals to obtain MM events, separated by at least 30 s, whereas in our study the total number of seconds for which the MM conditions are fulfilled is used. We expect that these different ways of assessing MM-like structures are still comparable on average. It is clear that these plots shows less structure than the middle panels of Figs. 7 and 8 because of the lesser amount of data and the coarser grid. We will compare a few of their conclusions with the results in the present study. Volwerk et al. (2016) state that:

1. The number of MM-like structures at solar maximum is higher than at solar minimum by $\sim 14\%$;

2. The probability is the same for solar minimum and maximum conditions;

3. The distribution of $\Delta B/B$ is exponential with approximately the same coefficient for both solar conditions;

4. For solar minimum the general trend for MM-like structures is to decay; for solar maximum MM-like structures first grow and then decay, between the bow shock and the terminator;

Point (1) is in general agreement with what is shown in Table 1, albeit that the increase for solar maximum is $\sim 45\%$, even though the total residence time for solar maximum was $\sim 10\%$ less. This again has influence on point (2), regarding the probability. In our present study we find that in total the probabilities of detection of MM-like structures are $\sim 0.05$ and $\sim 0.08$ for solar minimum and maximum (Table 1), respectively, i.e. a multiplication factor $\times 1.6$, between solar minimum and solar maximum conditions. This shows that considering a larger statistical data set for this kind of study greatly influences the statistical results.

Fig. 10 shows the distribution of $\Delta B/B$, which seems to indicate an exponential drop-off as in point (3). Volwerk et al. (2016) found two different e-folding lengths, $b$, through a fit by:

$$y = a \times \exp\{b \times \Delta B/B\}, \tag{7}$$

describing the distribution for "weak" ($\Delta B/B < 0.6$) and "strong" ($\Delta B/B > 0.4$) MM-like structures[4], with $b \approx -3.4$ and $b \approx -2.5$ respectively, for solar minimum and maximum conditions. In our present study the data seem to show also an exponential decay, however with three slopes (see Fig. 15): $b \approx -9.9$, $b \approx -5.4$, and $b \approx -10.5$. This significantly differs from earlier results.

---

[4]The depth of the MM-like structures has been adjusted to agree with the definition in this current paper.

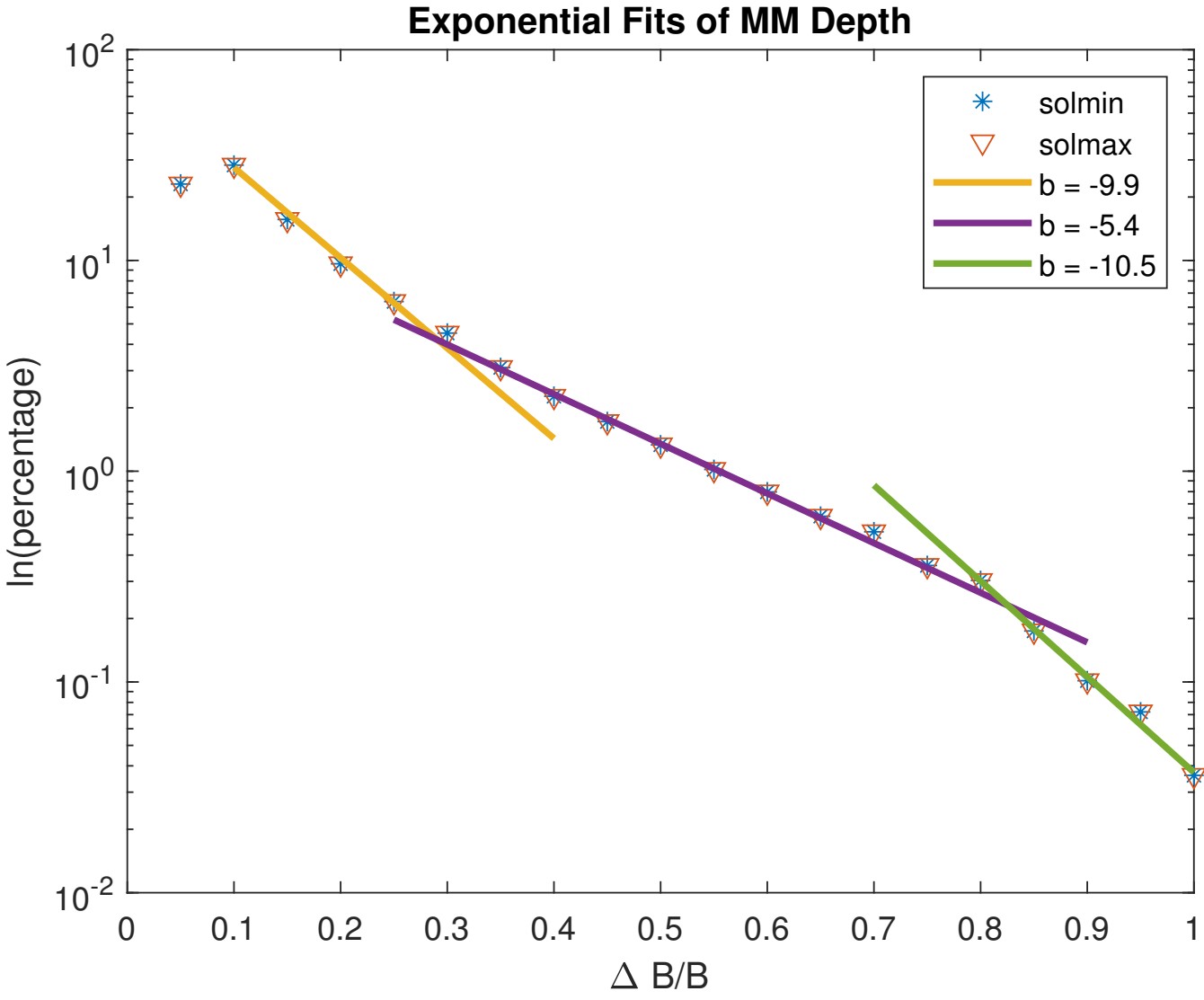

**Figure 15.** Exponential fits to the depth distribution of the MM-like structures for solar minimum and maximum. Three different regions can be identified and the slopes $b$ for the fits $y = a * \exp\{b \times \Delta B/B\}$ can be found in the legend.

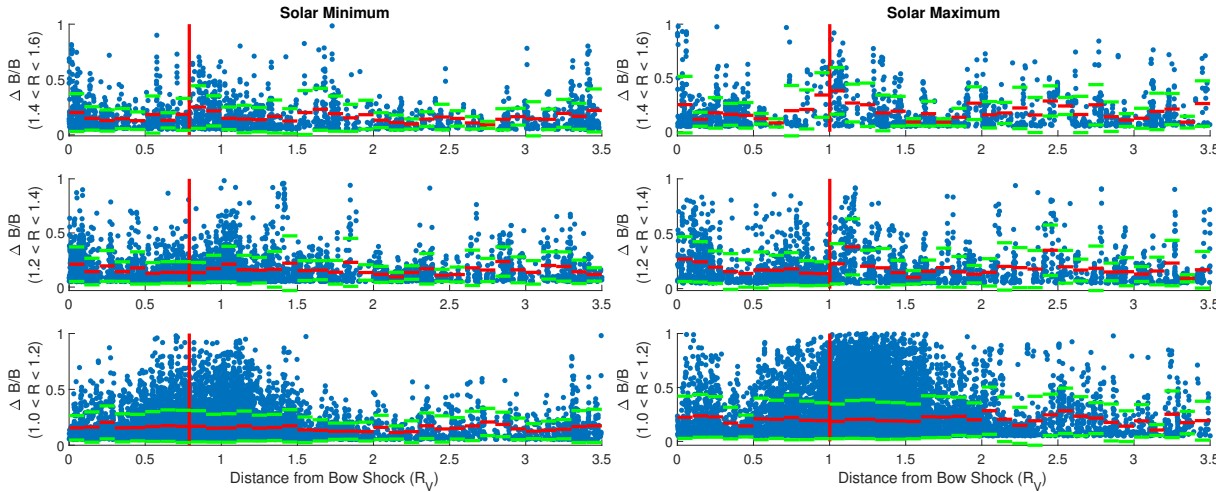

**Figure 16.** $\Delta B/B$ as a function of distance from the bow shock for three intervals $1.0 \leq R \leq 1.2$, $1.2 \leq R \leq 1.4$ and $1.4 \leq R \leq 1.6$. The red vertical line shows the distance of the terminator to the bow shock in the $1.0 \leq R \leq 1.5$ bin.

Looking at the distribution of the occurrence rate, and fitting the median and upper/lower-quartiles of $\Delta B/B$ taken in $0.25\,R_{\mathrm{V}}$-bins, Volwerk et al. (2016) found that for solar minimum there was a decrease of these numbers from the bow shock away, whereas for solar maximum these values were first increasing and then decreasing.

In Fig. 16, we first calculate the distance of any event on the map to the bow shock along the $X_{\mathrm{VSO}}$ line, with $R = \mathrm{constant}$. $\Delta B/B$ is plotted for three $R$-intervals. The vertical red lines show the average location of the terminator with respect to the shock. In the panels the mean (red) and standard deviation (green) are overplotted. There is an increase in the depth of the structures towards the terminator and slightly behind, although in the bottom panels for $1.0 \leq R \leq 1.2$ this is not quite visible in the mean and standard deviation.

This is different from Volwerk et al. (2016), where it was stated that the MM-like structures decay away from the BS at solar minimum and first grow and then decay at solar maximum. With better statistics, this is not the case, there is a drop in maximum $\Delta B/B$ from the BS inward, i.e. from 0 to $\sim 0.25\,R_{\mathrm{V}}$, for both situations.

A point not discussed in Volwerk et al. (2016), but well-investigated in Volwerk et al. (2008c), is the role of the bow shock characterisation. It is expected that the MMs are mainly generated behind a quasi-perpendicular bow shock (where the crossing ions are mainly heated in the perpendicular direction to the magnetic field) and close to the magnetic pile-up region close to the planet (through conservation of the first adiabatic invariant), from which they then are transported by the plasma flow. Figure 6 in Volwerk et al. (2008c) shows the occurrence rate in the dayside magnetosheath for various angles ranges between the IMF and the bow shock normal. It shows clearly that the occurrence rate drops significantly when the bow shock is quasi-parallel. Above in Fig. 12 a similar effect is observed. The split into three bins, quasi-perpendicular, intermediate and quasi-parallel, shows that for the latter, there are almost no events ($\sim 3\%$) and that most of the events are found in the intermediate group ($\sim 68\%$).

## 5.2 Comparison with Rojas Mata et al. (2022)

Lately, Rojas Mata et al. (2022) have studied the Ion Mass Analyzer (IMA) proton data of the ASPERA-4 instrument, showing the proton temperature anisotropies during the whole VEX mission, divided up into solar minimum and maximum, similar as in our present study. In Fig. 17 the temperature ratio $\mathcal{T} = \mathrm{median}(T_\perp/T_\parallel)$ for each grid cell is shown on a smaller grid $(0.1 \times 0.1\,R_\mathrm{V})$ than in the original paper. The overall trends are similar for both grid resolutions except a few random "outlier-looking" bins. It was found that the highest temperature anisotropy, $\mathcal{T}$, lies deeper inside the magnetosheath for solar maximum. Indeed, the same conclusion can be made from Fig. 17, where, like in the occurrence rate in Fig. 8, we have plotted the solar minimum location of the bow shock as a magenta dashed line. Interestingly, this line seems to lie well along the boundary of the maximum $\mathcal{T}$.

It should be noticed that there is a bias in the comparison between the ASPERA-4 data, with 192 s resolution and the magnetometer resolution of 1 s. The ion data will show larger local variations as with the orbital velocity of VEX of $\sim 8$ km/s this results in a size of $\sim 1500$ km or $\sim 0.25\,R_\mathrm{V}$ (the tentative reason to have this grid cell size in Rojas Mata et al. (2022)), although this does not take into account the (much) faster plasma flow in the magnetosheath. However, the averaging done in the grid cells might reduce this effect slightly in the statistics.

Comparing the $\mathcal{P}$ distributions in the middle panels of Figs. 7 and 8 with $\mathcal{T}$ in Fig. 17 one notices that, as said above, there are two regions of maximum $\mathcal{P}$, whereas the maximum of $\mathcal{T}$ seems to fall in-between these two regions. The effect is most clearly visible for solar maximum conditions. This shows that the presence of MM-like structures locally reduces the (median) temperature ratio $\mathcal{T}$ in the magnetosheath, an indication that the instability transfers its energy from the ions to the waves.

Figure 18 shows the percentage of scans in each $0.1 \times 0.1\,R_\mathrm{V}$ cell, for which the instability criterion $\mathrm{MMI} < 0$ (Eq. 1) is fulfilled. Note the large difference between the two solar conditions: although the criterion is fulfilled much more frequently at solar minimum than at solar maximum, there are fewer MM-like structures on average as shown in Figs. 7 and 8. This points at an extra necessity for MM-like structures to start to develop, apart from the instability criterion of Eq. (1)

## 6 Conclusions

We have studied the magnetic field data for the whole Venus Express mission and searched for mirror modes in the magnetosheath. The VEX mission has been split-up into two parts, corresponding to solar minimum and solar maximum, for which one of the main differences is that the bow shock for solar maximum is further out at $\sim 1.49\,R_\mathrm{V}$ as compared to $\sim 1.34\,R_\mathrm{V}$ for solar minimum at the subsolar point.

The total probability $\mathcal{P}$ for solar maximum lies higher than for solar minimum even when normalising to the total observation time for each condition. The regions where the MM-like structures are observed are behind the BS and near the magnetopause/ionopause. But behind the BS, the probability $\mathcal{P}$ peaks further inside the magnetosheath for solar maximum. Comparison with the proton data shows that $\mathcal{P}$ peaks where the temperature anisotropy, $\mathcal{T}$, is reduced, indicating that energy has been transferred from the ions to the waves.

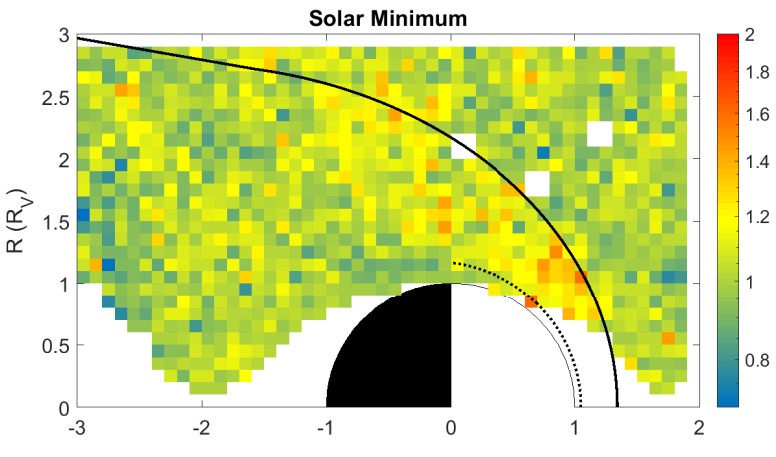

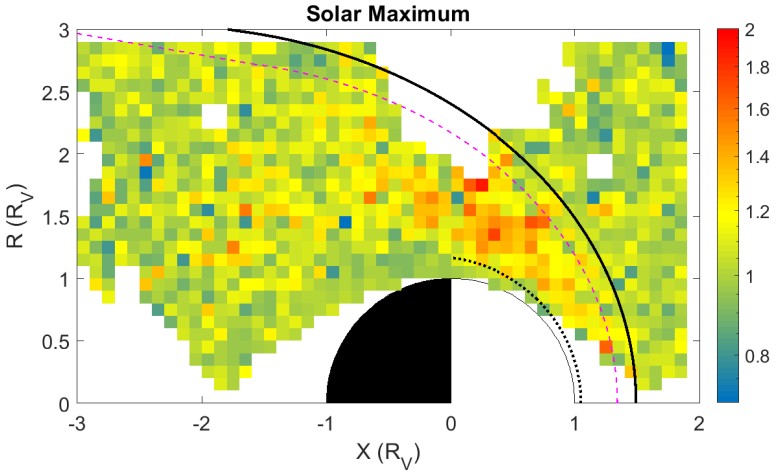

**Figure 17.** The temperature ratio $\mathcal{T} = \mathrm{median}(T_\perp / T_\parallel)$ for each grid box around Venus for solar minimum (top) and maximum (bottom). The data have been re-evaluated on a $0.1 \times 0.1\,R_V$ grid from the study by Rojas Mata et al. (2022).

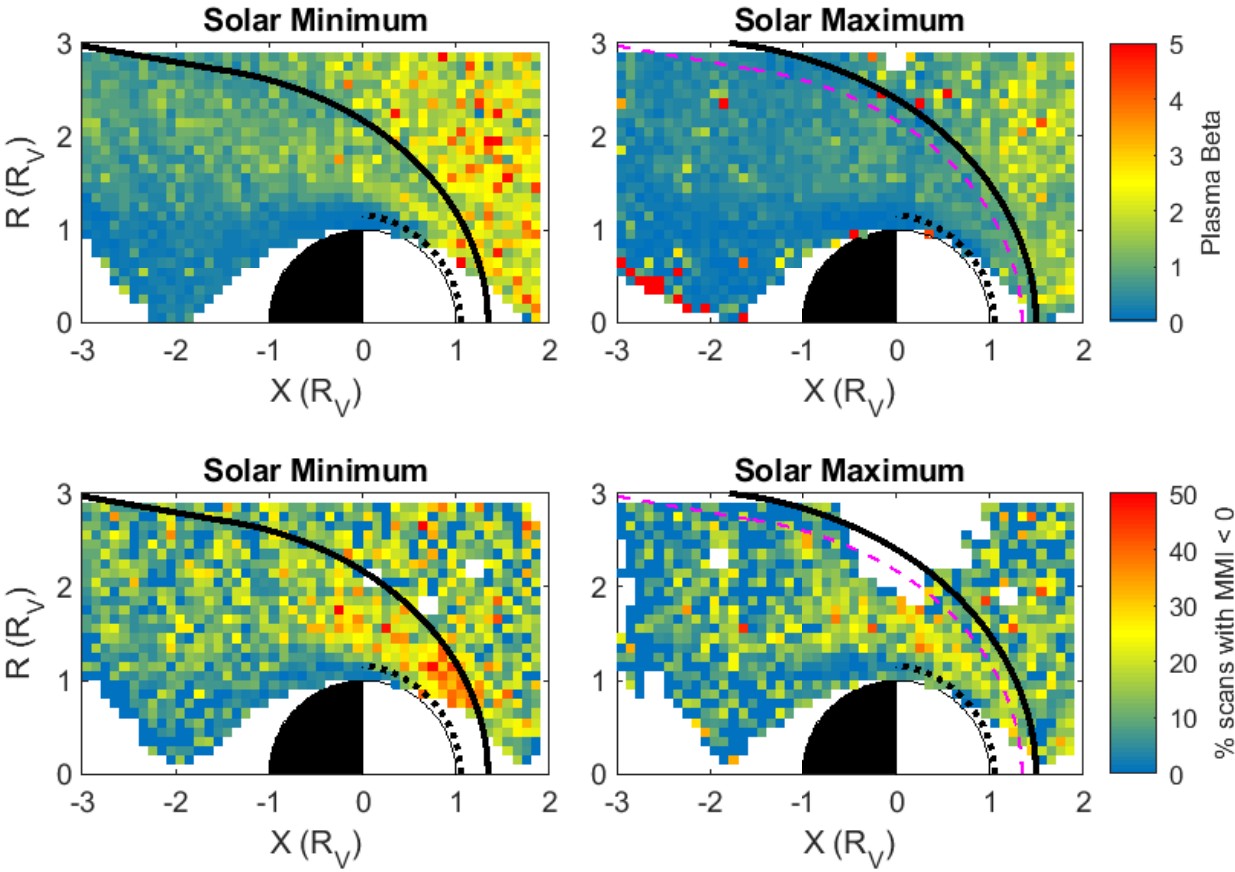

**Figure 18.** Top: The mean plasma-$\beta$ in Venus's environment. Bottom: The percentage of scans in each box for which the instability criterion, Eq. (1), $\mathrm{MMI} < 0$. The data have been re-evaluated on a $0.1 \times 0.1\,R_{\mathrm{V}}$ grid from the study by Rojas Mata et al. (2022).

The total probablity $\mathcal{P}$ is lower for solar miminum than for solar maximum (Table 1), but Fig. 18 shows that the percentage of scans with $\mathrm{MMI} < 0$ is larger for solar minimum. Again this raises the question whether there is more to the creation of MM-like structures than just the instability criterion. The major difference in the two periods is the plasma-$\beta$, shown in Fig. 18 (top panels) which is much higher for solar minimum (as also observed by Wilson et al., 2018, in the solar wind at 1 au). This extra thermal energy does not seem to drive more or even deeper MM-like structures as shown in Fig. 10. Quite possibly, there is a competing effect: Rojas Mata et al. (2022) show that there is a higher compression of the IMF during solar maximum in the dayside magnetosheath. This would decrease the plasma-$\beta$, but through the first adiabatic variant mechanism it could increase the perpendicular temperature and thereby increase the $\beta$ and the anisotropy. Indeed, in the magnetosheath behind the quasi-perpendicular bow shock an inverse correlation was found with variations between high-$\beta$ - low temperature anisotropy and low-$\beta$ - high temperature anisotropy (Anderson and Fuselier, 1993, 1994; Anderson et al., 1994; Fuselier et al., 1994).

The distribution of the depth of the MM-like structures does not seem to be strongly dependent on solar conditions. The distribution seems to be exponential, but closer inspection shows a combination of three different exponential slopes, with no difference between solar minimum and maximum. Also, solar irradiance, with proxy the F10.7 flux, does not seem to influence the number of MM-like structures.

There remain some open questions after this study. Why are there *large* regions where the temperature anisotropy, $\mathcal{T}$, is enhanced and the MM probability, $\mathcal{P}$, reduced? (This could be due to the difference between generation region and region of observation. Starting with a large anisotropy and creating MMs, by the time they are observed, the structures have already been transported downstream before they've had the chance to dissipate the free energy.) What are the extra conditions for MM-like structures to start to grow in the magnetosheath plasma? What is the dependence on solar wind IMF conditions? Also, the location of the solar minimum bow shock seems to take a special place, as during solar maximum it is the boundary where the probability, $\mathcal{P}$, strongly increases.

*Data availability.* The Venus Express magnetometer and IMA data are available through ESA's Planetary Science Archive (https://archives. esac.esa.int/psa). VEX Aspera-4 electron densities have been calculated from the respective electron spectra available in the ESA's PSA by M. Fraenz. The sunspot number was obtained from silso (http://sidc.be/silso/monthlyssnplot) and the F10.7 data were obtained from lisird (https://lasp.colorado.edu/lisird/data/penticton_radio_flux/

*Author contributions.* MV and CSW instigated the investigation. DM analysed the magnetometer data. MD calibrated the magnetometer data. SRM analysed the ASPERA-4 ion data. GSW, YF, CM, JH. DRC, and CB helped interpreting the results of the data analysis and in writing the paper.

*Competing interests.* There are no competing interests.

*Acknowledgements.* The work of CSW and DM is sponsored by the Austrian Science Fund (FWF) under project number P32035-N36. SRM was funded by the Swedish National Space Agency under contract 145/19 and 79/19. This research was supported by the International Space Science Institute (ISSI) in Bern, through ISSI International Team project #499 "Similarities and Differences in the Plasma at Comets and Mars." and through ISSI International Team project #517 "Towards a Unifying Model for Magnetic Depressions in Space Plasmas."

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
