# Peer review of "Statistical distribution of mirror mode-like structures in the magnetosheaths of unmagnetised planets: 2. Venus as observed by the Venus Express spacecraft"

_EGUsphere, 2022_

## Referee Comment (RC1)

**Review report, Volwerk et al., 'Statistical distribution of mirror mode-like structures in the magnetosheaths of unmagnetised planets: 2. Venus as observed by the Venus Express spacecraft'**

This paper reports on Venus Express magnetic field measurements of magnetic mirror mode waves in the magnetosheath of Venus. While this is a very interesting subject, unfortunately I have to recommend that this paper be rejected. I have two reasons for this:

1) There are actually very few new results in the paper. Almost all results are simply confirmations of earlier results reported by the first author [Volwerk et al., 2008a; 22008b; 2016]. In this manuscript, the whole Venus Express database is used for a statistical study. This of course has some value, but in my opinion does not in itself warrant a new publication. The authors indicate that they will use these data for a manuscript in which these results will be compared to similar data from Mars. This promises to be an interesting paper, and I would recommend to simply absorb the method and data from this manuscript into that paper.

2) Parts of the paper are rather poorly written, in particular the Introduction. There are also several misprints and mistakes that indicate that the manuscript has been put together in some haste.

Below I give some more detailed motivation for my recommendation.

*Introduction*:

The major flaw of the introduction is that it totally ignores on mentioning the work that has been already performed by the main author on this subject. That gives a somewhat misleading impression that the work presented in this paper is new and original, instead of putting it in context of earlier work. Comparison of earlier work only appears late in the paper (from line 265.)

lines 32, 40, 55, 58, (and also 311, 336, 352): The authors talk about '*temperature asymmetry*', while in reality they should be discussing temperature *anisotropy*.

line 35: a version of the magnetic mirror (MM) mode for multiple species is given (Equation 1). How is this relevant for this paper? What are the species to be used apart from protons? Cold electrons? Oxygen ions?

line 43: Here another form of an instability criterion is used, while discussing ion cyclotron waves. Why is the criterion given is this form? It is actually the same criterion as Eq. 1 for a single species, which is easily verified by a few lines of algebra. What is the relevance of giving this equation?

*Results:*

As mentioned, most results in this paper are simply confirmations of earlier results. Specifically:

Figure 2: Here even the same example as in [Volwerk et al., 2008a, 2008b, 2016] is used. Some new data would be in place here.

Figures 6-8: These are very similar to Figure 4 of [Volwerk et al., 2008b], and Figure 3 of [Volwerk et al., 2016].

Figure 11: Simply a reproduction of [Volwerk et al; 2016] (as the authors mention themselves.)

Figure 12: Very similar to Figure 4 of [Volwerk et al; 2016].

Figure 13: There is not much additional information as compared to Figures 5 and 6 of [Volwerk et al., 2016].

line 248-249: The conclusions here are the same as in the earlier work.

*General issues. These are just some examples of the paper not having been carefully prepared. I stopped taking notes at some point.*

line 46: '*B strength*'. Too informal.

line 47: '*conservation of first adiabatic invariant*'. Grammar.

line 47: '*The first process...*' This sentence discusses on of the processes responsible for creating a temperature anisotropy. Then the discussion ends abruptly with no further discussion of the other processes that are mentioned. Is is unclear what the authors want to say.

line 72: '*the dats*' ?

line 86: '*the data **B***' What is meant by this?

line 97: '*to which the reader is referred to*' Grammar.

line 101: $R_V$ not defined.

line 114: '*refsec:detection*' ?

line 137: What is the CSW method?

line 265: '*Comparison with Volwerk et al.*' Not correct reference format.

line 278: '*The probability for both solar conditions is the same for solar minimum and maximum*' Unintelligible sentence.

**References**

Volwerk, M., Zhang, T. L., Delva, M., Vörös, Z., Baumjohann, W., and Glassmeier, K.-H.: First identification of mirror mode waves in Venus' magnetosheath?, Geophys. Res. Lett, 35, L12204, https://doi.org/10.1029/2008GL033621, 2008a.

Volwerk, M., Zhang, T. L., Delva, M., Vörös, Z., Baumjohann, W., and Glassmeier, K.-H.: Mirror-mode-like structures in Venus' induced magnetosphere, J. Geophys. Res., 113, E00B16, https://doi.org/10.1029/2008JE003154, 2008b.

Volwerk, M., Schmid, D., Tsurutani, B. T., Delva, M., Plaschke, F., Narita, Y., Zhang, T. L., and Glassmeier, K.-H.: Mirror mode waves in Venus's magnetosheath: solar minimum vs. solar maximum, Ann. Geophys., 34, 1099 – 1108, https://doi.org/10.5194/angeo-34-1099-465 2016, 2016.

---

## Author Comment (AC1)

Reply to Referee 1

We thank the referee for taking the time and effort to evaluate our paper. Some of the points that the referee makes clearly need to be addressed. However, we do not agree that this is "the same paper as Volwerk et al. (2016)". This investigation has been performed on the full VEX data set, i.e. ~9 years, instead of 2 Venus years ~225 days (one during solar min and one during solar max), with newer and more stringent criteria than before. This was done to have the same analysis for Venus as was done for Mars in Paper 1, so in the end both planets can be directly compared. We do not think that is would be a good idea to change this paper directly into a comparison paper of Venus and Mars as that would lead to a very long paper. We are a bit surprised about the comments on some of the figures, that they are similar to those in Volwerk et al. (2016). It would be surprising if they were not, however with the much greater amount of data it was possible to decrease the grid size from 0.25 to 0.1 Venus radii. This shows much more details, which in the overall view might resemble the course figures, but which do give new and better information about the mirror mode structures around Venus. We do agree that some points could be presented better, which we will do in the revised version of this paper. Our replies to the comments of the referee are below.

The major flaw of the introduction is that it totally ignores on mentioning the work that has been already performed by the main author on this subject. That gives a somewhat misleading impression that the work presented in this paper is new and original, instead of putting it in context of earlier work. Comparison of earlier work only appears late in the paper (from line 265.)

The referee is correct, that the older paper should have been mentioned in the introduction, with the explanation why there is now a new, and more extended study of the MMs around Venus. This will be added to introduction.

lines 32, 40, 55, 58, (and also 311, 336, 352): The authors talk about 'temperature asymmetry', while in reality they should be discussing temperature anisotropy.

Indeed, there seems to be sudden change from "anisotropy" to "asymmetry". This is naturally incorrect, it should be "anisotropy".

line 35: a version of the magnetic mirror (MM) mode for multiple species is given (Equation 1). How is this relevant for this paper? What are the species to be used apart from protons? Cold electrons? Oxygen ions?

line 43: Here another form of an instability criterion is used, while discussing ion cyclotron waves. Why is the criterion given is this form? It is actually the same criterion as Eq. 1 for a single species, which is easily verified by a few lines of algebra. What is the relevance of giving this equation?

The general expression for the mirror mode instability criterion is given in the paper. However, it should be explained in the text that in the case that looked at in this paper only the protons are important. This then leads to another form of the instability criterion if some basic math is done, as shown in Eq. (3). This form of the instability criterion is given as it is used in recent papers.

Results:

As mentioned, most results in this paper are simply confirmations of earlier results. Specifically:

We disagree here with the referee, the much longer time that has now been investigated, i.e. the whole VEX mission instead of only 2 Venus years, shows that at least three of the conclusions in the Volwerk et al. (2016) paper, cannot be kept, because of statistics on a larger data set.

Figure 2: Here even the same example as in [Volwerk et al., 2008a, 2008b, 2016] is used. Some new data would be in place here.

True, the examples are the same, in order to show the difference in the more elaborate identification method that is being used now. This could have been shown with only one example, and new data should have been shown. One figure will be taken out and another figure, of MMs measured near the magnetic pile-up boundary will be included in the paper.

Figures 6-8: These are very similar to Figure 4 of [Volwerk et al., 2008b], and Figure 3 of [Volwerk et al., 2016].

They are similar but not the same, as here the full data set of the Venus Express mission is used and the grid size on which the statistics has been done has been reduced from 0.25 Rv to 0.1 Rv.

Figure 11: Simply a reproduction of [Volwerk et al; 2016] (as the authors mention themselves.)

True, this figure has been added as help to the reader. It needs not be in the paper, per se.

Figure 12: Very similar to Figure 4 of [Volwerk et al; 2016].

Naturally, this figure will look similar to the one in Volwerk et al. (2016). However, because of the larger dataset, it is now clear that there is not such a big spread between solar minimum and maximum, the asterisks and triangles basically lay on top of each other. Also, now there are three well defined slopes for the distribution of the number of events per depth of the events.

Figure 13: There is not much additional information as compared to Figures 5 and 6 of [Volwerk et al., 2016].

The referee must have a different view on the figures than we have. The perceived decrease (sol min) and increase (sol max) of the depth behind the bow shock is no longer apparent, nor that there is a difference between both solar activities.

line 248-249: The conclusions here are the same as in the earlier work.

True, but it needs to be stated here anyway that the much larger dataset statistically shows the same behaviour as in the older paper.

General issues. These are just some examples of the paper not having been carefully prepared. I stopped taking notes at some point.

line 46: 'B strength'. Too informal.

Will be changed in "slow changes in the magnitude of B"

line 47: 'conservation of first adiabatic invariant'. Grammar.

"the" will be added

line 47: 'The first process…' This sentence discusses on of the processes responsible for creating a temperature anisotropy. Then the discussion ends abruptly with no further discussion of the other processes that are mentioned. Is is unclear what the authors want to say.

There was a small paragraph missing here.

The first process will occur mainly in the solar wind interaction with the planetary exosphere (with the exception of Jupiter's magnetosphere and the Galilean moons, where the Jovian corotating magnetic field and magnetospheric plasma is taking the role of the solar wind) in the low-beta plasma case and generation of ion cyclotron waves will take place (Delva et al., 2008, 2009, 2011, 2015; Schmid et al., 2021). After crossing the quasi-perpendicular bow shock the anisotropy is increased, as is the plasma-beta and the MM instability will take over. The second process will occur mainly near the magnetic pile-up boundary where the magnetic field gets compressed and slowly increases in strength whilst getting closer to Venus.

line 72: 'the dats' ?

Typo: the data

line 86: 'the data B' What is meant by this?

The magnetic field data, B, are low-pass …

line 97: 'to which the reader is referred to' Grammar.

The second "to" is deleted.

line 101: RV not defined.

Omission: added "(Venus Radius, 1 R_V = 6051.8 km)"

line 114: 'refsec:detection' ?

LaTeX "\" missing.

line 137: What is the CSW method?

The "CSW method" abbreviation should have been mentioned in section 2.2. The two subsections 2.2.1 and 2.2.2 together build the CSW method, which is explained in more detail in Paper 1.

line 265: 'Comparison with Volwerk et al.' Not correct reference format.

The year has been added to both sections 4.1 and 4.2

line 278: 'The probability for both solar conditions is the same for solar minimum and maximum' Unintelligible sentence.

This is indeed doubled and should read "The probability is the same for solar minimum and maximum"

---

## Author Comment (AC2)

Reply to Referee 2

We thank the referee for taking the time and effort to evaluate our paper.

1. Mirror mode waves are generated by ion temperature anisotropies and usually identified by combining magnetic and ion observations. As shown by previous studies (Song 1994, Ruhunusiri 2015, Fraenz 2017, see below) most ULF waves in the Earth, Mars and Venus magnetosheaths are of Alfvenic type. ULF waves without field rotation are largely of fast Alfvenic type and only a small percentage are of mirror mode type. When using only a magnetic criterion one would expect that what is called 'mirror-mode-like' in the present study identifies mainly fast Alfvenic waves. Thus the term 'mirror-mode-like' is rather misleading.

This is a point of concern, naturally, when only using magnetic field data to determine which mode one is dealing with. However, since the presentation of the method by Lucek et al. (1999) there have been various studies which have confirmed that, indeed, mirror modes are detected with this technique, when plasma data are available (e.g. Rae et al., 2007 for events identified by Lucek et al. (1999), Volwerk et al. (2016) at comet 67P, Simon Wedlund et al. (2022b, Paper 1) at Mars). Therefore, we are rather confident that these strict requirements are leading to mirror modes. However, the lack of plasma data (but see a later comment below) lead us to call the structures mirror mode like. The nomenclature used is the same as in Paper 1, where Simon Wedlund extensively discusses why we use this.

Also, the Mars case study paper (Simon Wedlund et al. 2022a), discusses how good our detection method is wrt actual MMs and refined the criteria. Next to that our q-perp and q-para conditions' maps (not shown yet) are consistent with MM behaviour.

2. Why mirror mode occurrence should depend on solar activity is not clear. The proposed dependence on pick-up of exospheric ions is rather speculative. If this would rearly be the case a comparison between the occcurrence in Venus and Earth magnetosheath should be done. Since at magnetosheath location at Earth pick-up is very small a clear difference to Venus and Mars should be observed. More important than the pick-up could be the bow shock normal angle dependenece for the plasma downstream of the shock.

One reason for which we look at the effect of different solar activity on the occurrence rate of mirror modes is because of the strong difference in upstream ion cyclotron wave detection by Delva et al. (2015). This means increased ion pickup in the solar wind, and these ions will be transported through the bow shock into the magnetosheath. After the crossing of the quasi perpendicular bow shock the ions are energized again and then in a high-beta region will generate mirror modes. Whether this interpretation is correct or not, needed to be checked, even after Volwerk et al. (2016).

Probably, see also below, there will be a greater influence of the bow shock condition, i.e. quasi parallel or perpendicular, on the occurrence rate of the mirror modes. (see further below)

Comparison between Venus, Mars and Earth would be an interesting endeavour, and would fit in the planned paper 3, which would compare Venus and Mars.

In conclusion I recommend a major revision of the paper where

1. the results are discussed in relation to the more robust studies by (Song 1994, Ruhunusiri 2015, Fraenz 2017) and the term 'MM-like' should probably be replaced by fast Alfvenic.

We will extend the discussion with the references given by the referee. However, we do not think that replacing "MM-like" with "fast Alfvénic" is appropriate.

2. the physical influence of the pick-up process should be better proven or justified why it should have major influence only just behind the bow shock.

Naturally, an increased number of pick-up ions will not only have an effect just behind the bow shock. Delva et al. (2015) found an increase in cyclotron wave detections by a factor 3, caused mainly by an increased ion pick-up. It could be expected that this also has an influence on the generation of MMs in the magnetosheath. Indeed, we find that there is a small effect in the magnetosheath, with the probability of detecting MMs during solar maximum is 60% higher than for solar minimum.

Minor comments by line number:

32: it should be stressed that the theory of MMs usually only considers the ion temperature since the instability evolves on ion scales. The role of the electron temperature in this is less clear.

Eq. (1) shows the instability criterion for a plasma with all species *i* which include also the electrons. It has been shown, lately, that for small scale scale magnetic holes the electron vortex plays an important role (e.g. Wang et al., 2020). In order to investigate the effect of the electrons in Eq. (1) in Venus's magnetosheath, one would need the electron data from the ASPERA-ELS sensor. Fränz et al. (2017) show data from ELS where the electron time-energy plot gives an upper value of ~100 eV for the energy of the electrons in the magnetosheath and the density is of the order of ~10 cm-3. This can be used in comparing the influence of electrons as compared to the ions, under specific assumptions for the electrons (Tpar, Tperp).

50: it should be mentioned here which processes are relevant in the Venus magnetosheath.

Indeed, there was a part missing here. The paragraph should read:

The first process will occur mainly in the solar wind interaction with the planetary exosphere (with the exception of Jupiter's magnetosphere and the Galilean moons, where the Jovian corotating magnetic field and magnetospheric plasma is taking the role of the solar wind) in the low-beta plasma case and generation of ion cyclotron waves will take place (Delva et al., 2008, 2009, 2011, 2015; Schmid et al., 2021). After crossing the quasi-perpendicular bow shock the anisotropy is increased, as is the plasma-beta and the MM instability will take over. The second process will occur mainly near the magnetic pile-up boundary where the magnetic field gets compressed and slowly increases in strength whilst getting closer to Venus.

63: maximum

corrected

66: completely ignored in this introduction is a series of papers which analysis ULF plasma wave types using both magnetometer and ion spectrometer data and thus has a superior wave type identification:

   Song, Russell, Gary, JGR, 99,6011 (1994): Identification of low-frequency fluctuations in the terrestrial magnetosheath.

Ruhunusiri et al. GRL, DOI: 10.1002/2015GL064968 (2015): Low-frequency waves in the Martian magnetosphere and their response to upstream solar wind driving conditions

Fraenz et al., PSS, DOI: 10.1016/j.pss.2017.08.011 (2017): Ultra low frequency waves at Venus: Observations by the Venus Express spacecraft

Specifically the last paper applies the Song-Russell method to VEX magnetometer and ASPERA-4 ion and electron data to obtain a statistical wave type identification. Here the high temporal resolution (4s) of the electron spectrometer is used to obtain properties of the ion distribution under the assumption quasi-neutrality.

By this method the authors show that ULF waves of mirror mode type at the dayside of Venus occur only close to the MPB with a share of about 15% of all ULF wave types. We regard these results as much more robust than the results presented in the current manusccript.

The identification paper by Song et al. is an important method for identifying wave modes when one has a full set of plasma measurements at the appropriate sampling rate. This paper will be presented in the introduction, and then a reasoning will be given why we turned to the Lucek et al. method, because of missing plasma data at an appropriate resolution. Indeed, in the Fraenz et al. paper uses Venus Express ASPERA-4 ELS data to get a grip on the plasma characteristics, with various assumptions. There are some critical points in this method, e.g. the resampling of the data to 4 s resolution (to fit B and Ne), and it is not clear how this works out on the MVA, and the crude resolution used in Fourier analysis in 128 s windows may well fail to detect structures.
However, as mentioned above already, the robustness of the method used in this paper has been shown by other studies. Indeed, comparing the numbers in Fig. 6 middle panel one can see that the occurrence rates are actually less for the current study (<= 10%) than for the Fränz et al. study (~20 %).

Finally, it should also be mentioned that the Song-scheme has also been criticized many timed (e.g. Schwartz et al. 1996, Denton et al. 1998, and Sahraoui et al. 2003). Naturally, a qualitative comparison with the results from Fraenz et al. should be made, but quantitative comparison will be difficult because of the highly different detection methods.

75: yes, it is doubtful whether ASPERA-4 IMA data alone with 192s resolution and restricted field of view can be used in this context. Faenz et al. 2017 try to overcome this problem.

Indeed, the resolution of ASPERA-4 IMA is not enough to use for wave identification. We have now obtained the ASPERA-4 ELS data from Markus Fränz and can check with these the density variations for selected events.

85: Simon Wedlund (2022a) discusses that the identification of MM-like structures based on magnetometer data alone does not make much sense.

We think that the discussion by Simon Wedlund et al. (2022a) is more nuanced than the referee makes it sound. In that paper it is actually shown that the B-field only determination works well, checked later by using the plasma data on the MAVEN spacecraft. A further discussion of the validity of this method is given in Simon Wedlund et al. (2022b, Paper 1).

100: a bow shock model based on VEX observations was derived by Martinecz et al., JGR, DOI: 10.1029/2008JE003174 (2009) and later improved by Chai et al., JGR, DOI: 10.1002/2014JA19878 (2014).

Indeed, there are other bow shock models that can be used, as these two mentioned by the referee. We will mention these models in the paper.

113: section reference missing.

LaTeX typo

Figure.2: variation direction: It would be more interestin to see examples which also show MPB crossings because closer to the MPB usually more MM structures can be identified.

New examples will be added to the paper. Indeed, MMs near the MPB should also be shown, this was an omission in the paper.

The conclusion of this section seems to be that the CSW method is more accurate.

142: the definition of VSO is incorrect: Z_VSO points to Venus orbital North, not 'solar North'. It is not clear why the division into solar min and maximum is made. Physically it would make more sense to divide into observation behind a quasi-parallel and quasi-perpendicular bow shock.

Correct, Venus north and not solar north, this will be corrected.
The reason for looking at solar minimum and maximum has been discussed above already. However, the referee is correct that the effect of quasi perpendicular and parallel bow shock conditions should be more important, as shown e.g. in Volwerk et al. (2008b). The data have already been reprocessed with the predictor-estimator method for the bow shock location (Simon Wedlund et al., 2022a), where also the condition of the bow shock is determined. This will be analysed in the revised version of the paper.

155: it is confusing to discuss Fig. 7&8 at this point while they appear much later in the paper.

This is an editing error, after adding figures 4 and 5, these figures shifted down. They will be moved up again.

170: since no temperature data are used in this analysis the conclusion is pure speculation. If you compare with results by Fraenz 2017 it is found that the 'MM-like' structure behind the bow shock are just fast alfvenic waves.

We disagree, see discussion above about the detection methods.

190: also this discussion indicates that a separation of the data set according to upstream bow shock type would make more sense. The expansion of the exosphere during solar maximum causes mainly more ICWs.

See above.

202: is there any conclusion from Figs. 4&5? Otherwise they can also be omitted.

These figures show the various distributions of the characteristics of the MMs. A further discussion will be added to the paper.

Figure 6: what is the reference for the MPB or ionopause location?

The MP/Ionopause model is from Zhang et al. [2007]. The omitted reference will be added in the paper.

226: it is not clear why pick-up production should be higher just behind the bow shock. The hydrogen exosphere has an exponential fall in density independent of the bow shock.

Discussed above.

274: show

Corrected